# The role of elasticity on adhesion and clustering of neurons on soft surfaces
Giovanni Marinaro [1], Luigi Bruno [2], Noemi Pirillo[3], Maria Laura Coluccio [3], Marina Nanni[4], Natalia Malara [5], Edmondo Battista [6], Giulia Bruno[7], Francesco De Angelis[7], Laura Cancedda [4], Daniele Di Mascolo [8,9] ✉ & Francesco Gentile [3] ✉

The question of whether material stiffness enhances cell adhesion and clustering is still open to debate. Results from the literature are seemingly contradictory, with some reports illustrating that adhesion increases with surface stiffness and others suggesting that the performance of a system of cells is curbed by high values of elasticity. To address the role of elasticity as a regulator in neuronal cell adhesion and clustering, we investigated the topological characteristics of networks of neurons on polydimethylsiloxane (PDMS) surfaces - with values of elasticity (E) varying in the 0.55–2.65 MPa range. Results illustrate that, as elasticity increases, the number of neurons adhering on the surface decreases. Notably, the small-world coefficient – a topological measure of networks – also decreases. Numerical simulations and functional multi-calcium imaging experiments further indicated that the activity of neuronal cells on soft surfaces improves for decreasing E. Experimental findings are supported by a mathematical model, that explains adhesion and clustering of cells on soft materials as a function of few parameters - including the Young's modulus and roughness of the material. Overall, results indicate that – in the considered elasticity interval – increasing the compliance of a material improves adhesion, improves clustering, and enhances communication of neurons.

The biological functions of tissues and organs depend on the way cells interact with each other, send and receive signals, and exchange information[1–4]. The trafficking of electrochemical, mechanical or biological signals is influenced, in turn, by the layout and configuration of elements in a system: the efficiency of an organ is determined less by the characteristics of individual cells and more by the fact that a great many of these cells interact in systems with non-trivial topological properties[5–12].

Materials and interfaces are tools through which one can guide the organization of several different cells into systems with some kind of internal organization and structure[13–16]. Cells on a material surface sense—through transmembrane receptors—a variety of physicochemical, geometrical, and biological cues in response to which they regulate their functions, including adhesion, proliferation, and migration[13,17–19]. The balance between cell-surface and cell-cell interaction forces[20–26] can influence the collective motion of cells, their cooperation, the development and evolution of multi-cellular structures. For those involved in the design of biomaterials and scaffolds for tissue engineering and in-vitro-medicine applications is thus relevant understanding how the characteristics of a substrate influence cell behavior.

The relationship between the *geometry* of a surface at the nanoscale and cell behavior has been examined in a variety of studies[27–36]. Some of these have illustrated that surface roughness in the 20–40nm interval optimizes cell adhesion on silicon[37–39], mesoporous silicon[40–42] and zinc-oxide nanowire surfaces[43]. Possibly more important than pure adhesion, the same reports have highlighted that neurons on surfaces modified at the nanoscale assemble in *networks* with high clustering and short path lengths, and enhanced computational efficiency—compared to neurons uniformly distributed on a flat support, such as flat silicon or conventional Petri-dishes. While these and other similar reports have focused on the role of

[1]Center for Interdisciplinary Research on Medicines (CIRM), University of Liège, Quartier Hôpital, 4000 Liège, Belgium. [2]Department of Mechanical, Energy and Management Engineering, University of Calabria, 87036 Rende, Italy. [3]Nanotechnology Research Center, Department of Experimental and Clinical Medicine, University of "Magna Graecia" of Catanzaro, 88100 Catanzaro, Italy. [4]Department of Neuroscience and Brain Technologies, Italian Institute of Technology, Via Morego 30, 16163 Genoa, Italy. [5]Department of Health Science, University of "Magna Graecia" of Catanzaro, 88100 Catanzaro, Italy. [6]Department of Innovative Technologies in Medicine & Dentistry, University "G. d'Annunzio" Chieti-Pescara, 66100 Chieti, Italy. [7]Plasmon Nanotechnologies, Italian Institute of Technology, Via Morego 30, 16163 Genoa, Italy. [8]Laboratory of Nanotechnology for Precision Medicine, Italian Institute of Technology, 16163 Genoa, Italy. [9]Department of Electrical and Information Engineering, Polytechnic University of Bari, 70126 Bari, Italy. ✉e-mail: daniele.dimascolo@poliba.it; francesco.gentile@unicz.it

topography on the process of cell adhesion and clustering on a surface—they have, however, disregarded the mechanical characteristics of the substrate.

Other works have examined how substrate *stiffness* influences cell adhesion, cell migration and locomotion, and the formation of tissues and organs. Results of these studies are *seemingly* contradictory. While some works indicate that adhesion, clustering, and migration of cells is enhanced on stiff substrates[44–47], others—conversely—indicate that on soft substrates cells merge to form tissue-like structures[48]. In a recently published paper[49], Janmey and coworkers found that—contrary to the common belief that compliant surfaces are typically non-permissive for migration—cells from embryonic tissue dynamically decrease their stiffness in response to substrate stiffening thus triggering collective cell migration. Further to this end, in a comprehensive review of how the mechanical characteristics of the brain change following neuro-degenerative disorders[50], Hall and colleagues recalled that retinal ganglion cell axons develop in the direction of softer brain tissue. In the same review, it is also recalled that adult-born neurons mature more slowly in the aged brain - that is stiffer than the young - suggesting that the speed of neuronal maturation is related to the compliance of the neurogenic niche.

Notice though that these previous works have used different types of cells, such as fibroblasts[46], mammary epithelial cells[47], kidney epithelial cells[45], from different tissues, and these cells are highly differentiated, which may require a unique microenvironment for proliferation. Thus, the different behavior of cells (in terms of adhesion, proliferation, migration, differentiation) observed by different groups on soft or hard surfaces, may be ascribed to several factors other than stiffness, including the chemical structure of the material, cell type, the experimental or environmental conditions of the measurements.

In addition, notice that the cited, existing body of literature focused on relatively soft materials with elasticity in the $2 kPa$ to $65 kPa$ range. A highly cited review and a ref. 51 for those working in the field of biomaterials and regenerative medicine maintains that—in this elasticity range—while it is generally true that increasing substrate stiffness correlates with increasing cell differentiation, there are many exceptions, and the stiffness optimum for differentiation and other behaviors varies significantly from cell to cell. Further to this end, in another seminal work[52], it is reported that epithelial cells on soft gels (with $E \sim 1 kPa$) show diffuse and dynamic adhesion complexes; in contrast, stiff gels

(with E~100kPa) show cells with stable focal adhesions. In the same work, it is recalled that, in the low kPa range, soft deformable substrates enhance neurite branching. Thus, again, the lack of consensus on whether material stiffness enhances or undermines cell activity is, most likely, credited to the heterogeneity of conditions under which the great many of these studies have been performed.

Few studies have explored cell-surface interactions for relatively harder materials—with values of elasticity in the low MPa range. In ref. 53 it is illustrated that osteogenic differentiation and mineralisation by embryonic stem cells is enhanced on substrates higher Young's modulus (>2.3 MPa)—when compared to softer substrates with $E$ in the 0.04–1.9 MPa range. In another study[54], it is shown that polyurethane films with high values of Young's modulus (higher than approximately 4 MPa) enhance the adhesive capacity of NIH 3T3 fibroblasts and Wharton's jelly mesenchymal stem cells. The article presented in ref. 55 investigated how the properties of substrates influence the fate of stem cells. Researchers cultivated individual human epidermal stem cells on surfaces of polydimethylsiloxane (PDMS) and polyacrylamide (PAAm) hydrogels, varying in stiffness from 0.1kPa to 2.3 MPa, with collagen coating attached covalently. They observed that the stiffness of PDMS did not affect cell spreading and differentiation. Conversely, on low-stiffness PAAm (0.5kPa), cells failed to form stable focal adhesions and differentiated. Human mesenchymal stem cell differentiation was similarly independent of PDMS stiffness but was influenced by PAAm's elastic modulus. Analysis of dextran penetration revealed that less stiff PAAm substrates were more porous; suggesting that surface roughness can play a role during adhesion.

Our work fits into this long-standing series of previous studies. The aim of this research is elucidating how substrate elasticity—in the low 0.55–2.65 MPa range—influences neuronal growth, networking, and activity. To do this, we fabricated soft PDMS surfaces by replica molding techniques (Fig. 1a, b)—that were used as a substrate for neuronal cell culture and grow (Fig. 1c). Neuronal cells were then examined at 24, 48, 72, 96h from culture by fluorescence microscopy and also by functional multi calcium imaging (fMCI) (Fig. 1d). Fluorescence images of cells were processed using image analysis, network analysis, and information theory algorithms (Fig. 1e, f). Results of the analysis indicate that—in the considered elasticity range—adhesion and connectivity of neuronal cells are optimized for small values of elasticity of the substrate.

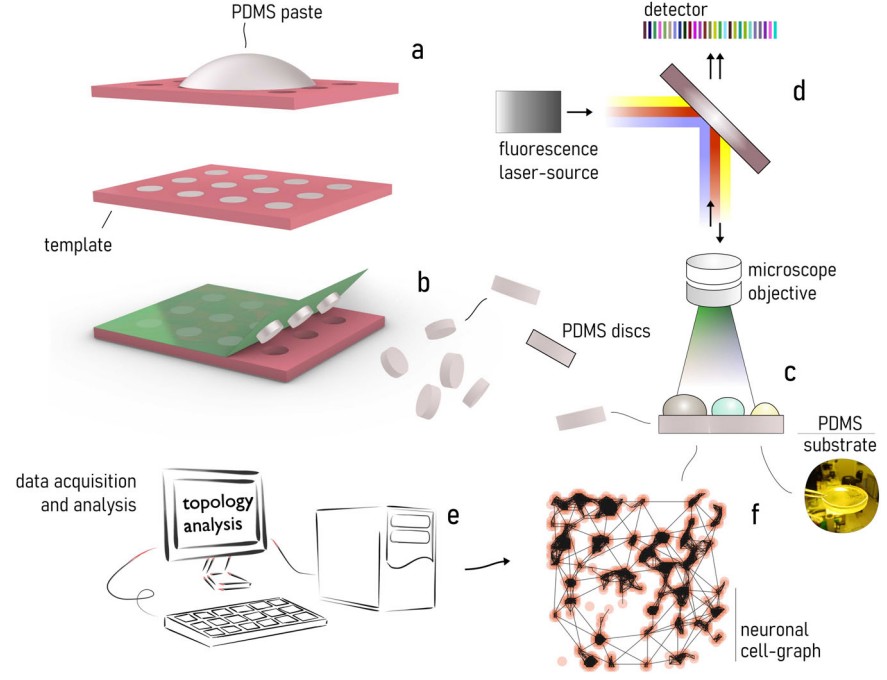

**Fig. 1 | Experimental set-up.** Using micromachining and replica molding techniques, we fabricated soft PDMS substrates for cell culture and growth. The elasticity of the substrates was varied in the $0.55 - 2.65 MPa$ interval (**a**). After detachment from the originating template (**b**) PDMS surfaces were incubated with primary neuronal cells (**c**) and placed on the stage of a fluorescence microscopy for investigation an**d** analysis (**d**). Fluorescence images of cells were processed using networks science, topology analysis, and information theory techniques (**e**). For each considered value of elasticity, we determined the topological attributes of neuronal cell networks forming on the substrates over time, made and estimate of the amount of information exchanged in the net, and measured cell activity (**f**).

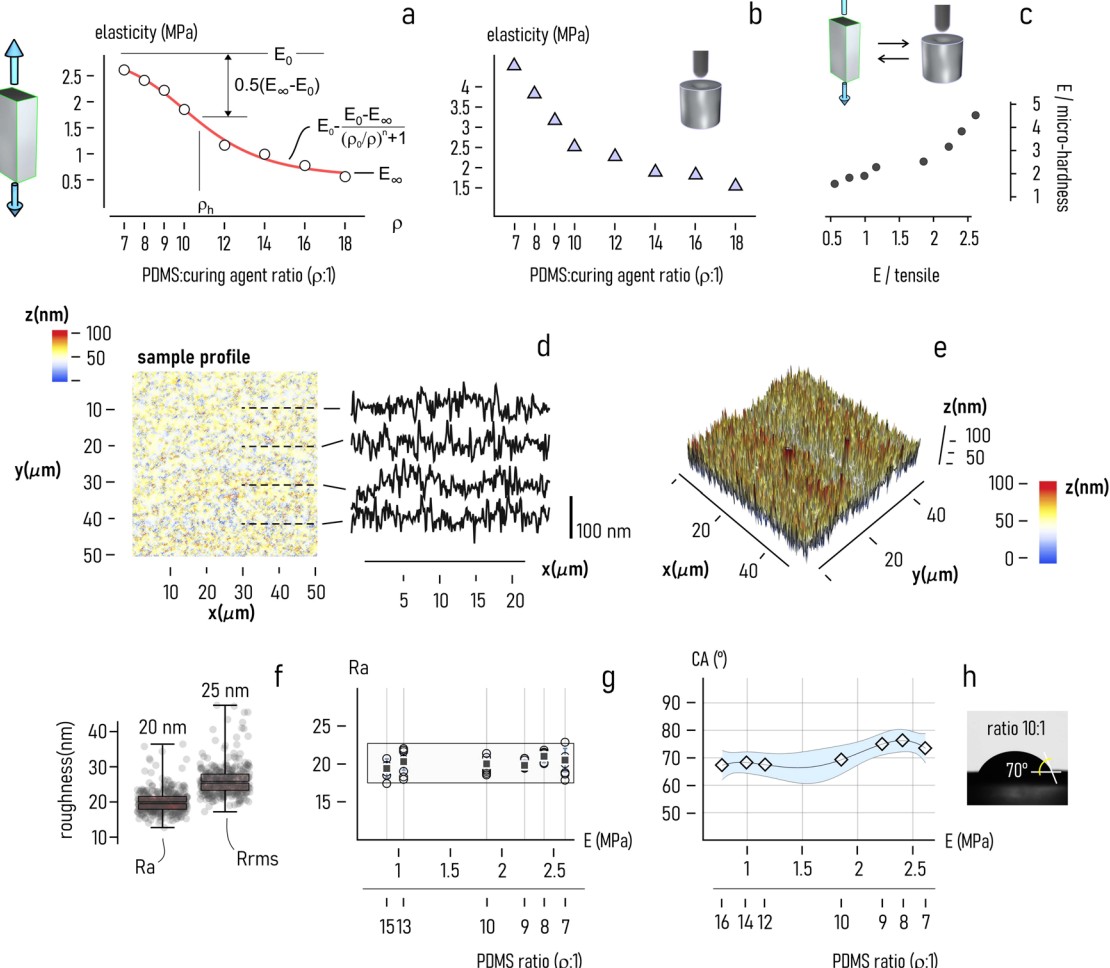

**Fig. 2 | Characterization of soft PDMS substrates.** PDMS surfaces were mechanically characterized using conventional elongation tests (**a**) and micro-indentation techniques (**b**) to find the elasticity of the substrate as a function of the ratio of liquid PDMS phase to the binding agent $r$: the larger the value of $r$, the smaller the value of elasticity. While results from the elongation test and micro-indentation of samples are consistent, however the latter overestimates the values of elasticity of a factor of 1.3 (**c**). Using laser interferometry, we measured the topography of PDMS surface represented here in the form of a linear and 2D density plot (**d**) and of a 3D plot (**e**). The average (Ra) and root mean square (Rrms) values of roughness were determined from morphological data. Values of Ra greater than 0 (Ra $\sim 20nm$)

evidence that at the nanoscale the PDMS surface is not flat (**f**). For different surface-preparations the values of Ra deviate marginally from the central value Ra $\sim 20nm$ (**g**). Contact angle (CA) measurements of samples indicate the PDMS surface is moderately hydrophilic with values of $CA<80°$ for all considered PDMS/curing agent ratios and values of elasticity $E<2.65MPa$ (**h**). Data in Fig. 2f are quantitatively described by a whisker box plot, where the lower and upper boundary corresponds to the 25% and 75% quartiles of the distribution, while the central band marks the median value (sample size $\sim 50$). Data in Fig. 2g are represented by mean ± standard deviation (sample size=10).

## Results

### Generating soft PDMS substrates

Using replica molding techniques described in the methods, we generated soft discoidal PDMS substrates with a diameter of $1cm$ and a thickness of $2mm$. The substrates were fabricated by mixing SYLGARD 184 elastomer (polymeric base, liquid) and curing agent material (polymerizing agent, solid phase) in different proportions $\rho$ that were varied between 7:1 and 18:1. The higher the content of polymeric base in solution, the larger the compliance of the resulting devices. We performed mechanical characterization tests to evaluate how the stiffness of the substrates correlates to the proportion between reagents in solution. After convenient sample preparation, PDMS specimens were characterized under uniaxial tensile loading using a universal testing machine (see the *methods*, the Supplementary Note 1 and Supplementary Fig. 1.1). In the tests, we used a loading rate of $1mN/s$—a sufficiently small value to ensure linearity. Other parameters of the tests are reported in the Supplementary Table 1. Resulting stress-strain relationships were processed (Supplementary Fig. 1.2,1.3) to derive the corresponding Young's modulus $E$ for each of the considered samples (Fig. 2a). Results

illustrate that the smaller the proportion of the polymeric base in solution, the higher the Young's modulus of the PDMS substrate. $E$ increases from about $E \sim 0.5MPa$ for $p = 18{:}1$ to $E \sim 2.5MPa$ for $\rho = 7{:}1$. However, the $E$-$\rho$ relationship is not linear, and for low ($\rho < 7$) and high ($\rho > 18$) values of $\rho$ the elasticity of PDMS reaches a steady state value. For this chemical formulation of PDMS, the elasticity of devices resulting from polymerization cannot be arbitrarily small or large. The best fit of the experimental data is an inverse logistic function of the form

$$E = E_o - \frac{E_o - E_\infty}{\left(\rho / \rho_h\right)^n - 1} \tag{1}$$

where $E_o$ ($E_\infty$) is the lower (upper) bound of $E$, $\rho_h$ is the full width at half maximum (FWHM) of the fitting curve, and $n$ is a model parameter (Fig. 2a). We calculated a $r^2$ statistics to examine whether the model template is consistent with the experimental data in the considered range of elasticity. Values of $r^2$ close to 1 ($r^2 = 0.998$) indicate that the predictions of the model match the observed values with high accuracy. For the data

considered in this study, we found that $E_o = 0.538 MPa$, $E_\infty = 2.809 MPa$, $\rho_h = 10.55$ and $n = 5.847$.

Additional micro-indentation testing of samples was performed to assess the reliability of mechanical sample characterization (Fig. 2b). Micro-indentation testing enables to estimate the mechanical characteristics of a sample by examining the impression left from the indenter in the sample material at the microscale (*methods* and Supplementary Note 1). Upon sample characterization, we found that values of elasticity predicted by micro-indentation and tensile loading tests are within 25% of each other (Fig. 2c). Results of the analysis are in line with previous reported studies[56] that indicate that micro-indentation techniques can possibly overestimate the value of the Young's modulus due to plasticity effects.

## Measuring PDMS surface characteristics

We used optical laser interferometry (*methods*) to measure the surface nanotopography of soft PDMS substrates. The surface profile measured over a region of $50\,\mu m \times 50\,\mu m$ is reported in Fig. 2d as a 2D density plot and in Fig. 2e as 3D plot for a sample device with $\rho = 10$ and $E = 1.88$ MPa. In the considered sampling area, surface height varies from $0\,nm$ (low) to about $100\,nm$ (high). One-dimensional scans along arbitrary directions were then isolated and independently analyzed to determine the average (Ra) and root square mean (Rrsm) values of roughness of the sample. Distributions of roughness values are described by the box and whisker plots reported in Fig. 2f, where the lower and upper boundary corresponds to the 25% and 75% quartiles of the distributions, while the central band marks the median value. Results of this descriptive statistics indicate that the mean values of Ra and Rrsm for the considered sample are $\langle Ra \rangle \sim 20\,nm$ and $\langle Rrms \rangle \sim 25\,nm$. As expected, values of Rrsm are systematically higher than Ra. Since values of Rrsm can be estimated from the Ra variable through simple values rescaling, in the following of the paper we will use the sole average roughness Ra to describe the morphology of sample surfaces. We measured surface topography of soft PDMS samples as a function of stiffness. Values of Ra are reported in the plot of Fig. 2g for 6 different sample configurations. Results indicate that for a sample stiffness varying between 0.88MPa ($\rho = 15$) and 2.65MPa ($\rho = 7$) the variation of surface roughness is negligible, with a mean value of $\langle Ra \rangle \sim 20.16\,nm$ and a small variations around the mean $\sigma(Ra) \sim 1\,nm$. Thus, in the considered elasticity interval the topography of soft PDMS substrates seems to be independent on sample preparation and is considered constant in the following of the paper.

Notably, the morphology of PDMS surfaces measured by Scanning Electron Microscopy (SEM) is consistent with the values of surface roughness determined by optical laser interferometry. SEM analysis was performed on PDMS sample surfaces with values of elasticity varying from 0.55 to 2.65 MPa (Supplementary Note 2). In all considered cases, surface profile measured by SEM is never perfectly flat (Supplementary Fig. 2.1). Instead, it always exhibits some spatial variability—consonant with the morphology measurement performed by quantitative interferometry analysis.

The characteristics of sample surfaces of being rough, with values of roughness in the $10 - 20\,nm$ range, can be accountable for the peculiar behavior of cells – that on soft PDMS substrates cluster into defined groups more markedly compared to harder substrates. The possible mechanisms and hypothesis underlying a similar behavior are explained in the following of the paper and in a separate supporting information.

The profile of PDMS surfaces was further processed. Morphological data of sample surfaces were Fourier transformed and the results of the transformation circularly averaged. The corresponding power spectrum (PS) density function (Supplementary Note 3 and Supplementary Fig. 3.1) describes in a logarithmic scale the change of information density of the sample surface per change of scale[38]. From the slope $b$ of the PS, one can thus determine the fractal dimension of sample surfaces as $D_f = (b - 8)/2$. For this configuration $D_f \sim 2.4$: strictly larger than the Euclidean dimension of bi-dimensional surface $D = 2$.

The wetting characteristics of the soft PDMS substrates were determined by measuring the contact angle $\vartheta$ of a drop of water deposited on the sample surface (*methods*). Measured values of $\vartheta$ vary between 67° ($E = 0.77$ MPa) and 76° ($E = 2.41$ MPa), substrates are thus moderately hydrophilic (Fig. 2h). The corresponding average surface energy density was then determined using the Young–Dupree equation as $\gamma \sim 96\,mJ/m^2$ with a variation smaller than 6%.

Since the non-specific energy of adhesion $\gamma$ determines the extent of adhesion of cells on a surface, the findings of our study that $\gamma$ is approximately constant suggests that the chemical properties of samples are not relevant to the conclusions of the research - and that the effects of stiffness and surface chemistry can be conveniently decoupled. However, to dissect even further whether the interaction between neuronal cells and the PDMS material may depend on other factors other than stiffness - we examined the chemical structure and composition of PDMS surfaces, and the interaction between PDMS and poly-d-lysine by Raman spectroscopy (Supplementary Fig. 4.1-4.5) and Energy Dispersive X-Ray Analysis (EDAX) (Supplementary Table 2). Results of these additional test campaigns—reported and conveniently commented in a separate Supplementary Note 4 - illustrate the polymeric-base/curing-agent ratio affects significantly material stiffness, and less significantly the chemical structure of samples. Thus, the results of this characterization seem to indicate that neuronal cell adhesion and clustering observed in this study, are significantly influenced by stiffness, while other factors, like chemical structure, are less relevant.

## Adhesion of neuronal cells on soft PDMS substrates

Soft PDMS samples were used as substrates for neuronal cell culture. Primary neuronal cells were incubated for a minimum of 24$h$ and a maximum of 96$h$ on sample surfaces with values of PDMS to curing agent ratio $\rho = 7 : 1$, $\rho = 10 : 1$, $\rho = 14 : 1$, $\rho = 18 : 1$, corresponding to values of elasticity $E \sim 2.65$ MPa, $E \sim 1.88$ MPa, $E \sim 1$ MPa, $E \sim 0.55$ MPa, respectively. Polystyrene rigid flat surfaces with a Young's modulus of $E \sim 3.4$ GPa were used as a control. At the end of the incubation period, cells were fixed, stained with DAPI (the nucleus) and imaged using fluorescence microscopy as described in the *methods*. The resulting fluorescence images illustrate that the number and network characteristics of neurons on PDMS substrates exhibit a very high sensitivity to substrate stiffness. In the considered range of stiffness (0.55 − 2.65 MPa), the lower the value of stiffness the higher the number of neurons adhering on the substrates. For sake of illustration, Fig. 3a compares fluorescent images of cell-nuclei taken 24$h$ from seeding on soft ($E = 0.55$ MPa, left) and rigid ($E = 1.88$ MPa, right) PDMS surfaces. Examples of fluorescent images of cells on PDMS with different values of elasticity, are reported in a separate Supplementary Note 5. Specifically, the Supplementary Fig. 5.1 reports examples of cell nuclei 24 $h$ from seeding on surfaces with elasticity varying from 0.55 to 2.65 MPa. The Supplementary Fig. 5.2 is a compilation of cell-nuclei images taken after a 24 $h$ incubation period on a $E = 0.55 MPa$ substrate. Similarly, Supplementary Fig. 5.3–5.5 report cell nuclei image-sets taken 24 $h$ from seeding on PDMS surfaces with value of elasticity ranging from 1 to 2.65 $MPa$.

Images of cells on the substrates at different values of elasticity and incubation time were processed using image analysis algorithms. For each configuration, cell-centers were identified and counted. More than 30 images were analysed per incubation time ($t$) and value of elasticity ($E$). Results of the analysis are reported in Fig. 3b–h.

After 24$h$ from incubation, the number of neurons ($N$) firmly adhering on a region of interest of $975 \times 750\,\mu m$ on the substrate is $N = 570 \pm 55$ for $E = 0.55 MPa$. On surfaces with value of elasticity $E = 1 MPa$, $N$ increases to $N = 719 \pm 79$. For values of stiffness greater than $1 MPa$ the number of cells on a region of interest steadily decreases with $E$, being $N = 544 \pm 22$ for $E = 1.88$ MPa and $N = 434 \pm 32$ for $E = 2.65$ MPa (Fig. 3b). For these data, the relationship between $N$ and $E$ can be approximated by a linear law of the type $N = E_o - \beta E$, where $E_o = 706$, $\beta = 92$ MPa$^{-1}$, and the values of $E$ are expressed in MPa (Fig. 3c). Thus, at 24$h$ and in the considered interval of values of the Young's modulus, the number of neuronal cells on soft PDMS surfaces decreases with $E$ with a rate of 92 cells per MPa. The same analysis, performed at 48, 72, 96 $h$ from incubation, reveals a similar decreasing trend of cell density with elasticity. For these times of the analysis:

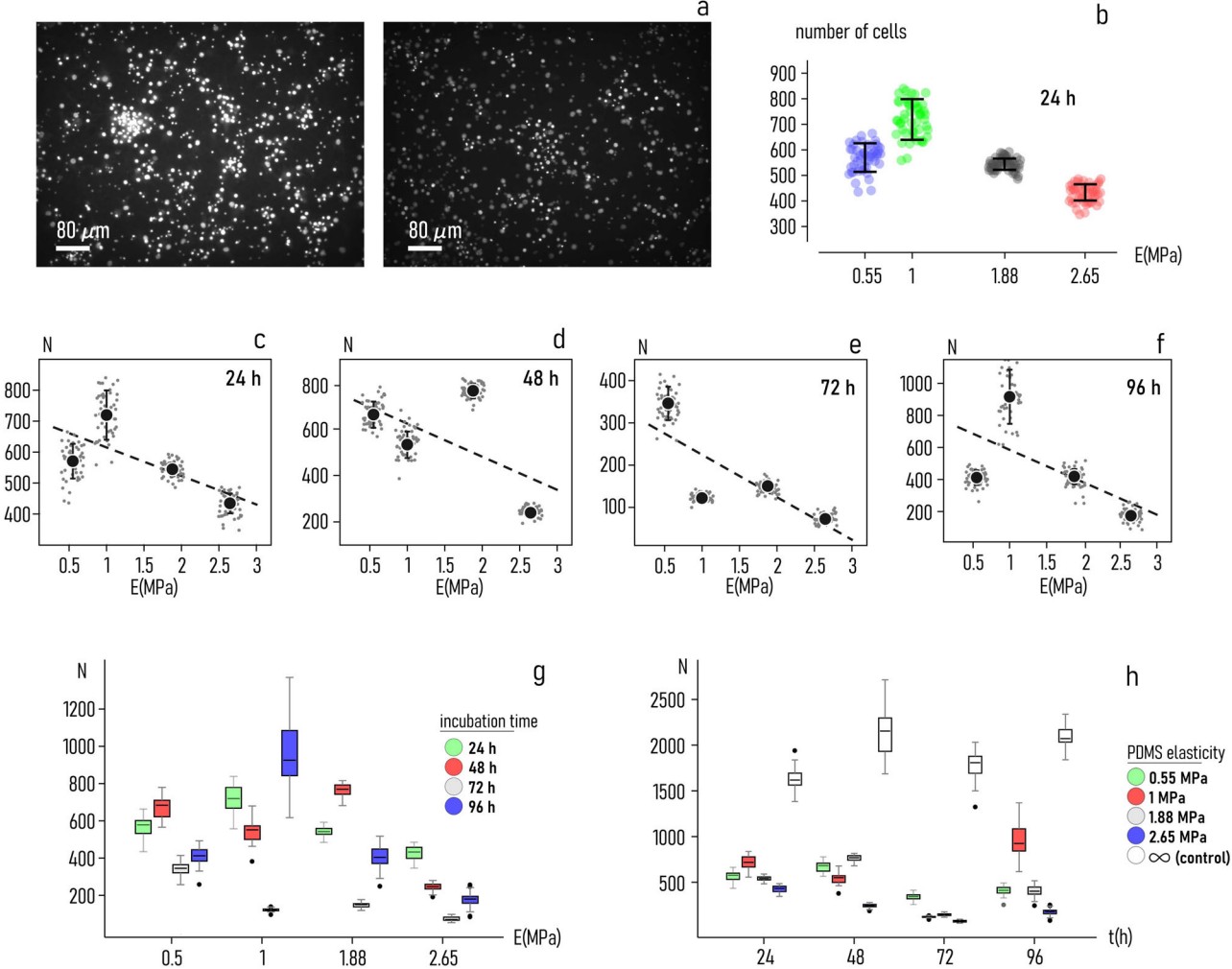

**Fig. 3 | Soft PDMS surfaces as substrates for neuronal cell growth.** Neuronal cells were plated on soft PDMS surfaces and followed over time. At fixed times, growth was stopped, cells immobilized and examined by fluorescence microscopy. Image shows how cell number and layout is affected by substrate elasticity: cell-growth is hampered on substrates with larger values of elasticity (1.88 *MPa*, rigth) compared to substrates with smaller ones (0.55 *MPa*, left) (**a**). The number of neuronal cells $N$ measured on substrates 24 *h* from incubation illustrates that $N$ shows a nearly inverse relationship with $E$ in the $0.55 - 2.65MPa$ interval (**b**). The negative correlation between N and E is exhibited for all considered times of incubation – 24 (**c**) 48 (**d**) 72 (**e**) and 96 *h* (**f**). Diagrams illustrate how the number of cells varies as a function of substrate elasticity (time) for all the times of the analysis (substrate elasticity) (**g**, **h**). Data in Fig. 3b–f are represented by mean ± standard deviation (sample size ∼ 50 for each data point). Data in Fig. 3g, h are quantitatively described by a whisker box plot, where the lower and upper boundary corresponds to the 25% and 75% quartiles of the distribution, while the central band marks the median value (sample size ∼ 50 for each data point).

$\beta = 142 \, \mathrm{MPa}^{-1}$ (48 *h*, Fig. 3d), $\beta = 100 \, \mathrm{MPa}^{-1}$ (72 *h*, Fig. 3e), $\beta = 202 \, \mathrm{MPa}^{-1}$ (96 *h*, Fig. 3f). Values of $N$ as a function of elasticity reported for all considered times in the same diagram (Fig. 3g) provide an overall view of results.

While the $N - E$ relationship is not perfectly linear, however in the considered elaticity interval the number of neurons *overall* decreases with the Young's modulus. For all the time points of the analysis, $N$ is always higher on softer ($E \sim 0.55MPa$) than on harder ($E \sim 2.65MPa$) surfaces. In the $0.55 - 2.65$ MPa range, the association between $N$ and $E$ is moderate to strong – and never weak – as evidenced by values of r-squared of $r^2 \sim 0.64$ for $t = 24 \, h$, $r^2 \sim 0.51$ for $t = 48 \, h$, $r^2 \sim 0.69$ for $t = 72 \, h$, $r^2 \sim 0.54$ for $t = 96 \, h$.

Values of cell density reported as a function of *time* ($t$) for different substrate-stiffness characteristics (Fig. 3h) illustrates the correlation between $N$ and $t$. For surfaces with elasticity $E = 0.55$ MPa, $N$ varies from $N \sim 570$ to $N \sim 412$ in the $24h - 96h$ interval. For $E = 1$ MPa and in the same interval of time, $N$ increases from $N \sim 718$ (24*h*) to $N \sim 917$ (96*h*). The number of cells decreases from $N \sim 543$ (24*h*) to $N \sim 418$ (96*h*) for $E = 1.88$ MPa; and from $N \sim 433$ to $N \sim 174$ for PDMS surfaces with $E = 2.65$ MPa. For comparison, the number of neuronal cells cultured on standard polystyrene

petri dish, used as a control, varies in the $1610 - 2100$ cells interval (Fig. 3h). The better adhesion performance of Petri dishes can be justified by the fact that these substrates are normally treated for improving cell attachment, while for these analyses PDMS substrates were not.

Best fit of experimental data with a linear model indicates that the rate of change of the number of cells is approximately $-3cells/day$ for $E = 0.55MPa$ ($r^2 = 0.88$), less than $\sim 1cells/day$ for $E = 1MPa$ ($r^2 = 0.84$). It is about $-4cells/day$ for $E = 1.88MPa$ ($r^2 = 0.82$) and $E = 2.65MPa$ ($r^2 = 0.94$). The rate of change of $N$ with time is vanishingly small, consistent with the notion that primary neurons in culture do not proliferate.

Notably, values of $N$ fluctuate for all substrate preparations between 72 and 96 h: data in Fig. 3 show a substantial decrease in the number of cells at 72 hours followed by an increase at 96 hours. The observed variation might depend on statistical fluctuations in the experiment and the measurements. Notice though the same variation from a lower (at 72 h) to a higher (at 96 h) number of cells - is observed in the control, i.e. substrates with theoretically infinite elasticity. Thus, the oscillation of $N$ measured for all substrate types may simply reflect an initial unbalanced distribution of cells during cell seeding.

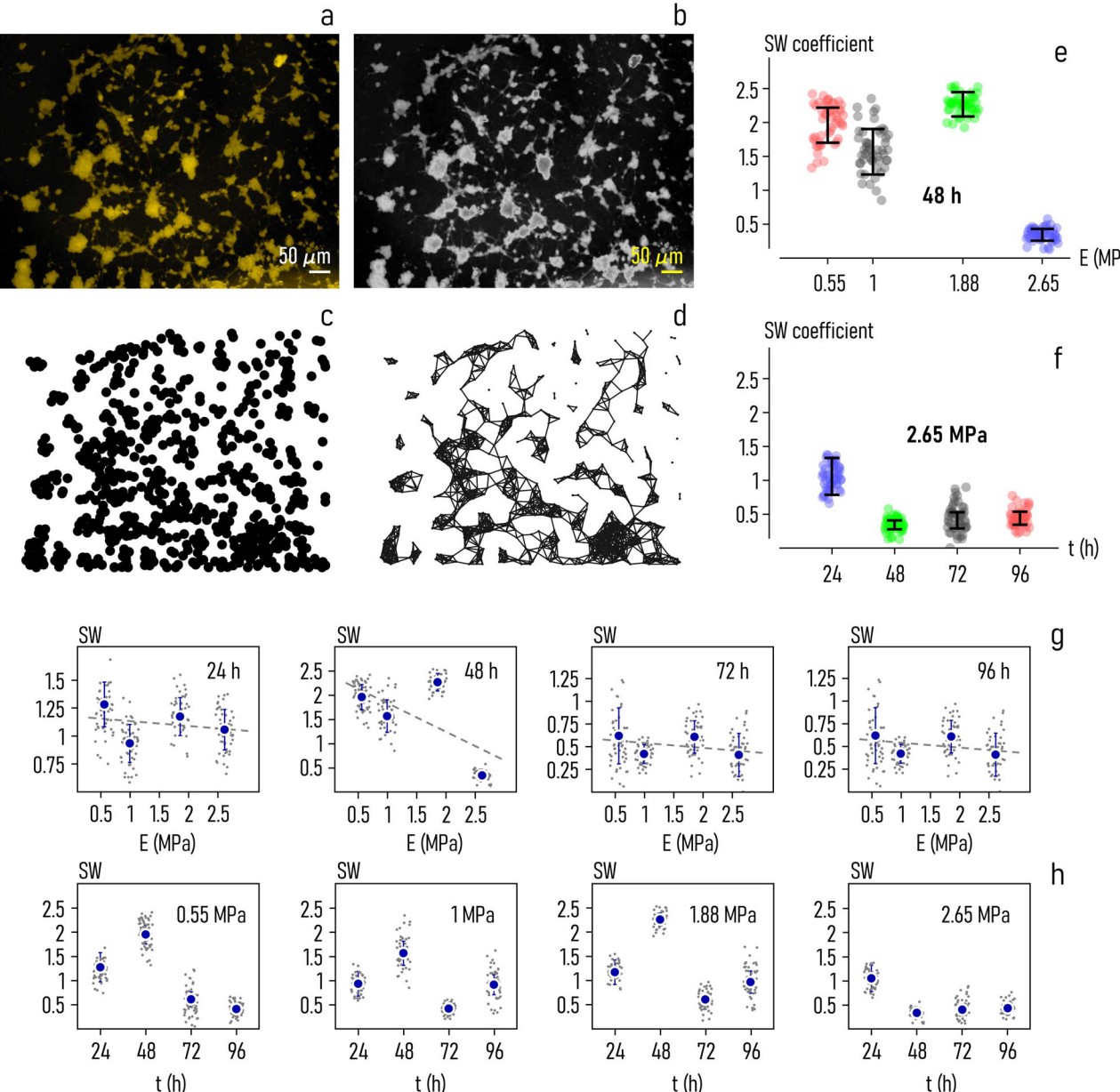

**Fig. 4 | Networks of neuronal cells on soft PDMS surfaces.** Visual examination of fluorescence images of neuronal cells suggests that surface stiffness can influence cell-clustering (**a**). We used image-analysis algorithms and networks-science to examine quantitatively the topological characteristics of cell-networks on the substrates. Fluorescence images of cells were gray-scale converted (**b**) and processed to extract cell-centers (**c**). Then, cell-centers were linked using the Waxman algorithm and a density-based rule (**d**). The small world coefficient (SW, a topological measure of networks) of neuronal-cell graphs as a function of surface elasticity, determined 24 h from culture – the diagram suggests that the ability of cells to form structured networks decreases with $E$ (**e**). The small world coefficient of neuronal-cell graphs as a function of time for a fixed value of the Young's modulus $E = 2.65 MPa$ (**f**). Correlation between the SW coefficient of neuronal cell networks forming on a substrate and the substrate elasticity, for different values of culture time: 24, 48, 72, 96 h (**g**). Correlation between the SW coefficient and time, for different values of substrate elasticity: 0.55, 1, 1.88, 2.65 MPa (**h**). Data in Fig. 4e–h are represented by mean ± standard deviation (sample size ∼ 50 for each data point).

## Small-world characteristics of neuronal cell networks on PDMS surfaces

We used networks analysis to examine how cells cluster together as a function of the mechanical characteristics of the surfaces. Fluorescent images of cells on PDMS substrates (Fig. 4a), were processed (Fig. 4b) using image analysis algorithms recalled in the methods of the paper. This allowed extraction of features such as the coordinates of the centers of the cells (Fig. 4c), that were in turn connected to generate neuronal cell-networks (Fig. 4d). A network is mathematical representation of elements in a space: it provides information on how elements are connected and on how they interact[57]. A comprehensive collection of graphs derived from fluorescent images of cells on PDMS surfaces at different values of elasticity, is reported in a separate Supplementary Note 6. Supplementary Fig. from 6.1–6.5 report examples of networks determined from neuronal cells imaged 48h from culture for values of substrate elasticity ranging in the $0.55 - 2.65 MPa$ interval.

In this study, cells were connected using a mixed distance and density rule (*methods*). We advanced the hypothesis that cells of the system are linked either if they are placed at a relatively short distance from each other or if they exhibit high local density and high distance from other cells with higher density—typical of cluster centers[58].

Once that cell-networks were deduced for each sample, we calculated their topological characteristics, including the small-world coefficient (SW). A network is *small-world* (SW>1) if it exhibits shorter paths and higher

clustering than an equivalent random graph with the same number of nodes and links, for which $SW = 1$[59–61]. The small-world coefficient is a measure of the shape and performance of a network—networks with SW>1 are morphologically structured to transport information efficiently[62,63].

Network analysis of cell-graphs indicates that the small-world coefficient of neuronal cell networks decreases with substrate stiffness in the $0.5 - 2.65$ MPa interval. This is especially evident $48h$ from culture (Fig. 4e). At this time of measurement, the value of SW transitions from $SW = 1.95 \pm 0.26$ for $E = 0.55$ MPa to $SW = 0.34 \pm 0.12$, for $E = 2.65$ MPa. For intermediate values of elasticity: $SW = 1.57 \pm 0.33$ for $E = 1$ MPa, and $SW = 2.26 \pm 0.179$ for $E = 1.88$ MPa. In this range of elasticity, SW varies with $E$ as $SW = 2.42 - 0.59E$.

We observe that the small-world coefficient of cell networks drops with $E$ also at 24, 72, 96 $h$ from culture, although the variation may be somehow less pronounced (Fig. 4g). $24h$ from seeding, SW decreases with $E$ following the linear law $SW = 1.18 - 0.04E$. After $72h$ and $96h$ of incubation, we found that relationship between SW and $E$ is approximated by the functions: $SW = 0.59 - 0.05E$ ($72h$) and $SW = 0.71 - 0.05E$ ($96h$). For comparison, the small-world coefficient of on neuronal cell networks on polystyrene, used as a control, is $SW \sim 1$ for all the considered time intervals.

*The dependence of SW on time* is explained by the diagrams in Fig. 4f and h. For cells cultured on 2.65MPa rigid surface, we found that SW decreases smoothly with time, moving from an initial value $SW \sim 1$ measured at $24h$, to a steady state value $SW \sim 0.44$ reached at $96h$ (Fig. 4f). Results from surfaces with other values of elasticity show the same trend (Fig. 4h). For $0.55$ MPa soft surface, the small-world-ness of neuronal networks shifts from $SW \sim 1.3$ measured at $24h$ to $SW \sim 0.42$ at $96h$. For surfaces with $E = 1$ MPa and $E = 1.88$ MPa, best fit of experimental data with a linear model yields a rate of variation of SW with time of $\beta = 0.12$ day$^{-1}$ for 1 MPa and $\beta = 0.22$ day$^{-1}$ for $E = 1.88$ MPa.

Neuronal cell networks were determined for a value of connection probability—a parameter in the wiring model—of $p = 0.92$ (*methods*). To demonstrate that results are robust against variations of the model parameters and that the $SW - E$ dependence is an underlying characteristic of neuronal cells, we performed additional analysis where $p$ was smoothly varied in the $0.84 - 0.96$ range.

Results of the analysis, reported in a separate Supplementary Note 7, illustrate that the values of SW decrease with $E$ for all the considered probabilities $p$ (Supplementary Fig. 7.2, 7.4)—with the exception of $p = 0.96$ at $24h$ (Supplementary Fig. 7.1), and $p = 0.86$, $p = 0.88$ and $p = 0.96$ (Supplementary Fig. 7.3) at $72h$ from culture. Thus results of this extended simulation campaign mostly support the findings of this section.

Results of this section need to be commented even further. In this study—the topological characteristics of neuronal networks have been determined starting from the positions of the nuclei on the surface—that have been then elaborated through convenient wiring models. However, this represents just an estimate of how cells connect. A similar model based on cell-cell-distance and cell-density, may explain less neuronal cell connectivity and more cell-condensation and clustering. To generate more reliable neuronal cell networks and provide a consolidated reference against which results of the work can be verified, we performed additional analysis. In place of examining cell-nuclei, we analyzed neuronal branching from green fluorescent images of cells, in which actin filaments were labelled using green fluorescent staining phalloidin conjugate (Supplementary Figs. 8.1–8.3). Since actin filaments are expressed in subcellular structures such as growth cones or dendritic spines, they can be used to dissect neurite outgrowth or synapse physiology. Results of this independent analysis (reported in a separate Supplementary Note 8) illustrate that the small-world coefficient of neuronal cell graphs decreases linearly with the Young's modulus as $SW = 1.98 - 0.62E$ (Supplementary Fig. 8.4). In contrast, the relationship found by wiring models of cell nuclei is (at $48h$ from culture) $SW = 2.42 - 0.59E$. The very close resemblance between linear model fits obtained using different procedures indicates that the main findings of this research study based on a mixed distance and density rule wiring model of cells – are accurate.

Specifically, that the small-world characteristics of neuronal networks are hindered by surface stiffness in the low MPa range.

## Statistical analysis of results

Analysis of variance (ANOVA) was used to examine whether the differences between SW means measured on substrates with varying elasticity are statistically significant. For samples measured $24h$ from analysis, ANOVA results indicate that networks cultured on surfaces with Young's modulus $E = 0.55 MPa$ exhibit a SW coefficient that is a different from the control ($E \to \infty$, $SW = 1$) at a significance level $\alpha = 0.01$ (Fig. 5a). The same test indicates that the differences between the $E = 1.88 MPa$ group and the control are significant at a level $\alpha = 0.05$ (Bonferrroni post hoc test conducted on the whole dataset).

$48h$ from culture, Bonferrroni post hoc test indicates that the differences among sample means and the control are statistically significant for each considered value of elasticity at a significance level $\alpha = 0.01$. $72h$ from culture, the small-world ness of networks on PDMS surfaces diverges in all cases from the control at a significance level $\alpha = 0.01$. After $96h$ from the beginning of experiment, substrates on which cells are different from the control have values of elasticity $E = 0.55 MPa$ and $E = 2.65 MPa$ ($\alpha = 0.01$).

## Estimating the amount information exchanged in neuronal cell networks

Since networks-analysis results indicate that surface elasticity influences the topology of neuronal cell graphs, we performed further research to examine whether the shape of networks (SW) affects in turn the ability of networks to exchange signals.

We used simulations (*methods*) to determine how information is transported in neuronal-cell networks as a function of substrate preparation. We followed the propagation of an electrical disturbance applied to a node of a system of neurons. Because of the stimulus, the node is subjected to a progressive increase of the potential across its membrane. When the value of the membrane potential exceeds a threshold, the neuron generates an action potential that is transported downstream the network, stimulating other neurons of the grid. The sequence of action potentials measured in correspondence of individual nodes of the network represents a signal. The information content of these signals—proportional to the rate and frequency of neuronal firing[64]. can be decoded using information theory approaches[65–67] and is measured in bits. Here, we first simulated the information transported in networks of neurons (Fig. 5b, c) and, secondly, determined the total information as the sum of individual pieces of information processed by nodes of the network. We determined the value of total information in neuronal-networks cultured on soft PDMS substrates with values of elasticity falling in the $0.55 - 2.65$ MPa interval. The networks of neuronal cells were built from images taken $24h$ from incubation. To assure statistical significance, we performed more than 10 simulations per neuronal network. Results of the analysis (Fig. 5d) illustrate that the average value of information is $I_{0.55} = 4.6 \pm 0.38 bits$ for cells adhering on a surface with Young's modulus $E = 0.55$ MPa. The value of information plunges to $I_1 = 3.8 \pm 0.47 bits$ for cells on a surface with $E = 1$ MPa. For values of elasticity moving from $E = 1.88$ MPa to $E = 2.65$ MPa, the information transported in networks of neuronal cells decreases from $I_{1.88} = 3.5 \pm 0.3$ to $I_{2.65} = 3.3 \pm 0.38 bits$. In the considered elasticity interval, the more rigid the PDMS substrates the less efficient the neuronal cell network developing on those surfaces.

## Measuring spontaneous neuronal cell activity

We used fMCI (functional Multi Calcium Imaging) techniques to measure the spontaneous activity of neuronal cells in networks cultured on PDMS surfaces with varying compliance. Samples were prepared following the protocols reported in refs. 39,43 and the methods. Then, substrates with subconfluent clusters of neuronal cells were imaged at DIV 15 using a fluorescent upright microscope. In each cell body, we recorded the intensity of fluorescence over time, $F(t)$, and determined the *variation* of fluorescence as

**Fig. 5 | Statistical analysis and simulations.** We used analysis of Variance (ANOVA) test to compare the small-world-ness of networks formed on different surfaces. Multiple-comparison post hoc Bonferroni test (a series of t-tests performed on each pair of groups corrected by the number of groups) indicates which samples means are significantly different from the control, i.e. random neuronal-networks cultured on rigid surfaces with SW = 1. In the diagram, sample-means that are different at some significance level $\alpha$, are marked by a bar. If $\alpha$ is less than 0.05, it is flagged with 1 star (*). If $\alpha$ is less than 0.01, it is flagged with 2 stars (**) (**a**). We used information theory to estimate the amount of information transported in networks of neuronal cells. We built connected graphs from fluorescence images of cells on the substrates (**b**) and examined how an initial disturbance propagates in those networks - resulting space and time patterns of signals were used to estimate the information processed over time in each node (**c**). Results of this theoretical analysis: total information $I$ elaborated in neuronal cell graphs cultured on soft PDMS surfaces, as a function of surface elasticity (**d**). Data in Fig. 5a and d are represented by mean ± standard deviation. The sample size for data reported in Fig. 5d is 10.

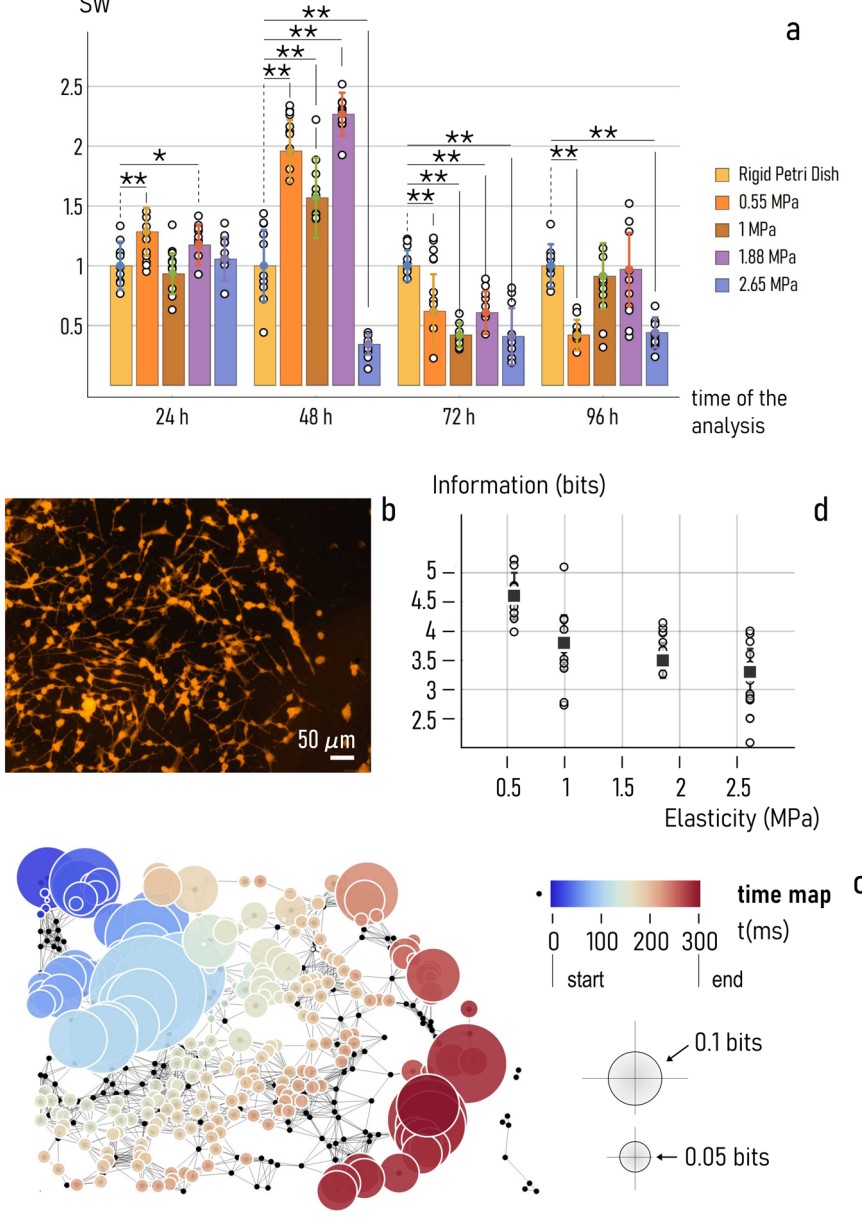

$\Delta F/F_o$, where $\Delta F = F - F_o$ and $F_o$ is the baseline[68]. The action potentials released in the system were determined as the onsets of individual $Ca^{2+}$ transients[68,69]. Figure 6a and b are fluorescence images of neuronal cells acquired at consecutive intervals showing how electric signals pass through the network. For networks cultured on substrates with 3 different values of elasticity - 0.55, 1 and $2.65 MPa$ - we registered the signal over time on several different sites (neurons) of cell-networks. Figure 6c shows the intensity of fluorescence measured over 6 separate neurons in neuronal networks cultured on PDMS surfaces with 0.55, 1 and $2.65 MPa$ Young's modulus. For all tracked neurons, fluorescence signals were registered for $200s$ and baseline corrected. Diagrams in Fig. 6c illustrate that the number and density of peaks and the neural activity vary with the elasticity of the substrate - the larger the Young's modulus, the less the abundance of peaks in each dataset. We then converted these analog signals into binary time series of 0/1 digits, where 1 (0) denotes the presence (absence) of activity. In this analog-to-digital conversion we used $\delta F = 0.5$ as cut-off value. The raster-plot representation of these discrete spiking events enables direct visualization of neuronal cell activity as a function of the surface characteristics and allows quantification of cell performance (Fig. 6d). For each

dataset acquired on substrates with 0.55, 1 and $2.65 MPa$ value of Young's modulus, we calculated the *frequency* of peaks as the number of peaks measured in a time interval, divided the length of the interval. Results of the analysis are reported in the diagram in Fig. 6e. The average firing frequency measured in the networks of neuronal cells is $f_{0.55} = 12.8 \pm 3.6$, $f_1 = 8.4 \pm 2.2$, $f_{2.65} = 2.0 \pm 0.45$ spikes/s for substrates with Young's moduli of 0.55, 1, and $2.65 MPa$, respectively. Moving from $E \sim 0.55$ to $E \sim 2.65 MPa$, neuronal cell frequency decreases of more than 6 times, indicating that – in the considered elasticity range – neuronal cell activity improves on substrates with higher compliance.

**Ruling out the role of PDMS leakage in determining cell-behavior**
To examine whether results of the work and the peculiar cell behavior that we observed are ascribable to toxicity effects of uncured PDMS material rather than to elasticity, we have performed an additional test campaign. The campaign was aimed at characterizing the leakage of PDMS into DI water at different times and under different temperatures and was carried out using both Raman spectroscopy and energy dispersive X-ray spectroscopy (EDX) techniques (Supplementary Fig. 9.1). PDMS substrates,

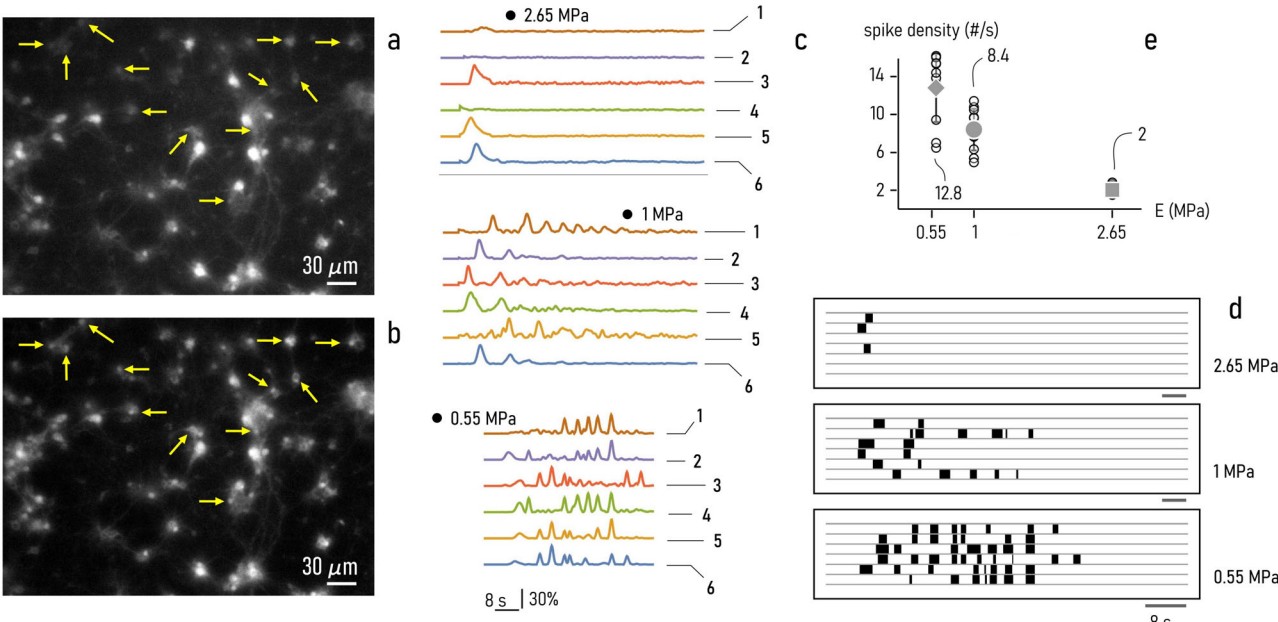

**Fig. 6 | Measuring neuronal-cell activity.** We used fMCI (functional multicalcium imaging) to measure the activity of neuronal cells on soft PDMS substrates. In the technique, calcium ions within neuronal cell networks are selectively targeted with a fluorescent compound - its transients (**a**, **b**) are then associated to the generation and release of action potentials in the system. Intensity of calcium-related fluorescence vs time measured at 6 different sites (neurons) of neuronal networks cultured on substrates with decreasing values of elasticity: 0.55, 1 and 2.65 MPa (**c**). Raster-plot of fluorescence intensity signals shown in **c**, **d**. Density of peaks of the fluorescence-intensity signals measured in neuronal-networks as a function of substrate elasticity (**e**). Data in Fig. 6e are represented by mean ± standard deviation (sample size = 12).

molded into cylindrical shapes with a diameter of $2\,cm$ and a height of $0.5\,cm$ using varying ratios of the liquid PDMS to curing agent ratio ($\rho$:1, $\rho = 7, 10, 13, 15, 18$), were submerged in $200\,\mu l$ of water to assess their stability. The testing was conducted at two different temperatures: 37 °C, to mimic cell culture conditions, and 60 °C, to accelerate the leaching process. Additionally, the substrates were incubated for a time varying from $24\,h$ (a short incubation period) to $108\,h$ (a long incubation period). To analyze any products that may have been released following the long-standing exposition of PDMS to water, a drop from each solution was placed in duplicate on a CaF2 slide and on a CaF2 slide coated with a sputtered gold layer for Raman analysis, and on a clean standard SEM pin stub for EDX analysis, then allowed to dry. Samples were subsequently examined using a Renishaw inVia Raman microscope equipped with a 50× objective of a Leica microscope, and by a FESEM ULTRA-PLUS equipped with an SE2 detector.

Results of the analysis, reported in a separate Supplementary Note 9, collectively indicate that the leakage of PDMS may be less relevant than the mechanical properties of substrates in determining cell behavior, for reasons that can be summarized as follows: (i) The Raman signal of PDMS traces is, in any case, vanishingly small (Supplementary Fig. 9.2, 9.3), and could be detected only through SERS effects (Supplementary Fig. 9.4). This indicates that leakage of PDMS is negligible. (ii) The variation of Raman signal associated to Si-C stretching is small for varying values of $\rho$, that in turn indicates that the liquid-base:curing-agent ratio influences only moderately leakage (Supplementary Figs. 9.5, 9.6). (iii) EDX analysis of samples illustrates that, in a given amount of PDMS excess, the relative abundance of Silicon correlates poorly with $\rho$—similarly to other elements found in solution (Supplementary Figs. 9.7–9.13). This indicates that, when present, the effects of leakage cannot explain the enhanced adhesion and enhanced clustering of neurons—that is instead related to the inverse of $\rho$. (iv) Even assuming a significant release of Si into water or the culture medium used for neurons, silicon, in the form of silicon dioxide, or silicon-based nano- and micro-particles, and particulate, is generally considered to be biocompatible and not toxic to cells under many conditions[70,71]. The biocompatibility of Si-based nano- and micro-scale materials has been a focus of ongoing research efforts to understand the factors affecting their interactions with biological systems. Considering all

this, we confidently rule out that leakage is responsible of the peculiar cell behavior observed and reported in our work, either alone or combined with other mechanisms or PDMS characteristics, such as elasticity.

## Discussion and conclusions

Results of the study indicate that, in the low MPa range, increasing compliance enhances the interaction of neuronal cells with soft materials - that in turn translates into augmented adhesion, augmented clustering, and improved cell activity. For substrates with a modulus of Young ($E$) moving from 0.55 to about 2.5 MPa, the number of cells measured on a region of interest of approximately $1\,mm^2$ decreases steadily at a rate of ~92 cells MPa$^{-1}$ 24 $h$ from seeding. Cell-density falls even more steeply with $E$ at later adhesion stages – with a cell depletion rate as high as ~150 and ~200 cells MPa$^{-1}$, 48 and 96 $h$ from incubation. In the same elasticity range, the small world coefficient – a measure of how efficiently cells form structured graphs with high clustering and short paths – also decreases. For networks measured 48 $h$ from seeding, SW transitions from SW~2 ($E = 0.55$ MPa) to SW~0.3 ($E = 2.65$ MPa). The increased ability of cells to form clustered networks at low values of elasticity has, as a result, an increased potential of those networks to process information – as evidenced by computer simulations and functional multicalcium experiments. With the first predicting information in low-elasticity substrates ($E = 0.55$ MPa) some 1.5 times higher than in high-elasticity supports ($E = 2.65$ MPa) - and the latter illustrating that the activity of neurons on soft substrates is 6 times higher than on rigid surfaces. Here, we assume that neuronal activity is encoded by the number of action potentials in the time interval. To put results into context, in experiments where we examined how surface nanotopography modulates neuronal cell interaction[39,43], we found that in the 0-40 nm range the small-world-ness of neuronal networks increases by a factor of ~3, the simulated information by a factor of ~2.6, and neuronal activity of ~4 times. Thus in the considered dimensional intervals, the mechanical properties of a material are as important as material's morphology in regulating the behavior of neuronal cells.

Results of the study are counterintuitive. They illustrate that cell adhesion, cell clustering, and activity, are optimized on materials with low

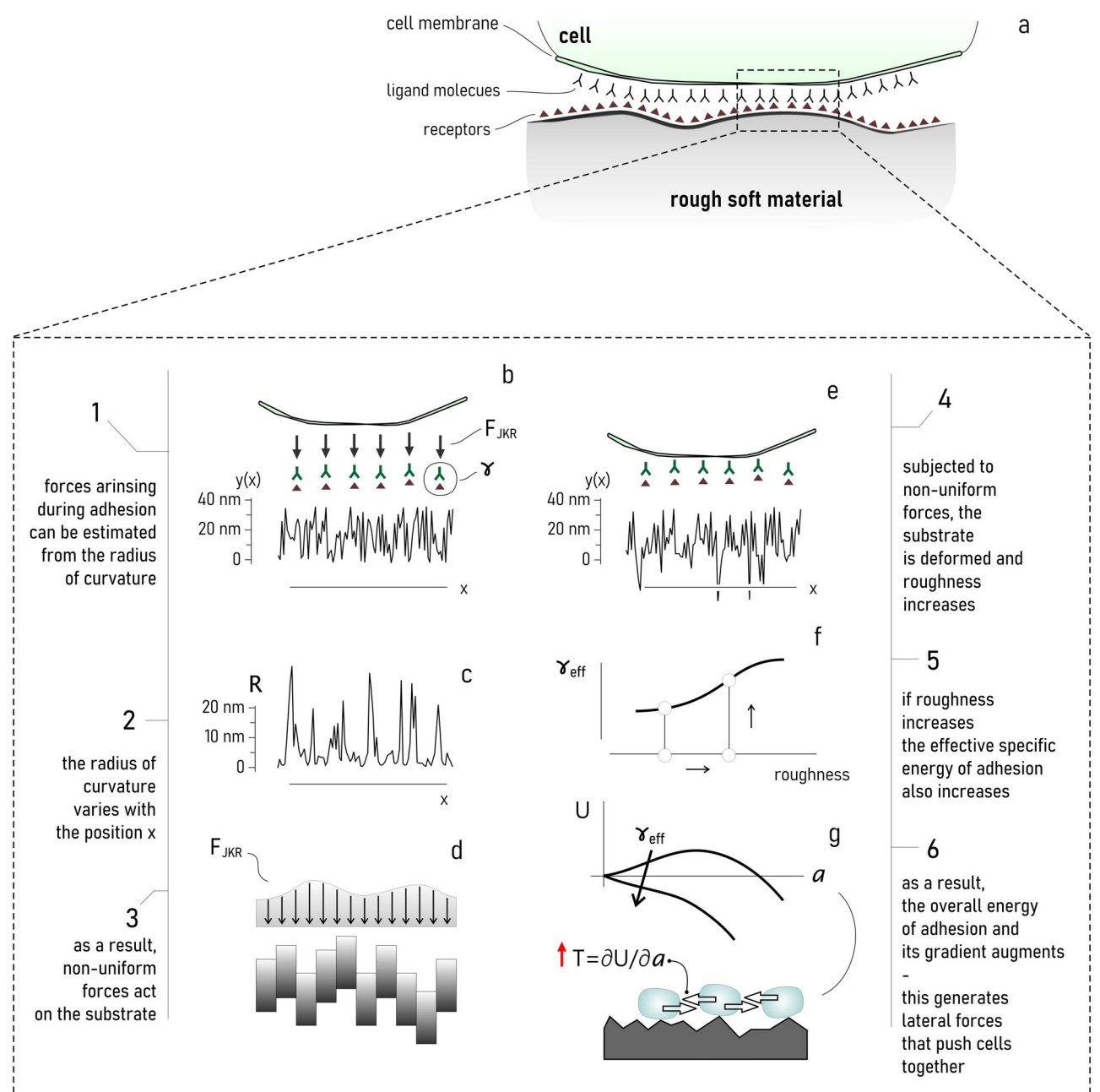

**Fig. 7 | Understanding the mechanisms of cell adhesion and clustering on soft materials.** Schematics of the process of adhesion of a cell to rough soft materials (**a**). The adhesion of a cell to a surface is mediated by cell-adhesion molecules and steric interactions with a specific energy of adhesion $\gamma$. Following the Johnson, Kendall and Roberts model, cell-membrane adhesive forces ($F_{JKR}$) depend on $\gamma$ and the radius of curvature ($\chi$) of the su**b**strate (**b**). Since PDMS material is a non-flat surface, $\chi$ is not uniform on the substrate (**c**). As a result, following adhesion PDMS surface is loaded with non-uniform forces distributed irregularly on the surface (**d**). Nonuniform forces deform unevenly the PDMS surface exacerbating the original roughness (**e**). An increased value of roughness has, as a consequence, the increase of the overall energy density of adhesion at the cell-surface interface (**f**). An enhanced adhesion generates in turn, as a collateral effect, lateral forces in the system that perturb cell-equilibrium and cause cell aggregation and clustering (**g**).

elasticity, contrary to the consolidated hypothesis that stiffness improves adhesion. But soft-material enhanced cell-adhesion may indeed be the case. In support to the experiments, we have developed a *simple* theoretical scheme that explains experimental data. Consider the scheme in Fig. 7a and Supplementary Fig. 10.1. The interplay between the cell and the surface is mediated by cell-adhesion molecules (CAMs) with chemical potential $c$, and by non-specific interaction forces described by a surface energy-density of adhesion $\gamma_{ns}$. Such that the overall specific energy of adhesion is $\gamma = c + \gamma_{ns}$. With Johnson, Kendall and Roberts[72], cells push on the substrate exerting an adhesive force $F = 3/2\ \pi\ R\gamma$, where $R$ is the local radius of curvature, and $\gamma$

coincides with the work of adhesion at the equilibrium (Fig. 7b). The radius of curvature can be determined as

$$R(x) = \left(1 + \left(\frac{dy}{dx}\right)^2\right)^{3/2} \Big/ \left|\frac{d^2y}{dx^2}\right| \qquad (2)$$

where $y(x)$ is the profile topography function of the position on the substrate $x$. Since the surface is not flat, i.e. $y$ is not constant (Supplementary Fig. 10.2), the radius of curvature is discontinuous with values that oscillate between

zero and a maximum (Fig. 7c and Supplementary Fig. 10.3). As a result, also the forces on the surface are irregular. Since the substrate is soft, it deforms under external loads and – assuming it is discretized into $n$ non-interacting elements – points of the surface are displaced downward of a quantity $\delta y(x) = y(x)F(x)/(E\,A)$, where $A$ is the section of the element (Fig. 7d and Supplementary Fig. 10.4). The final shape of the substrate is thus $y_{fin} = y_{in}(1 - F(x)/E\,A)$: for non-regular distributions of force and sufficiently large $\gamma$, $y_{fin}$ may be also significantly different from $y_{in}$ (Fig. 7e). This implies that, following cell-adhesion, substrate roughness increases (Supplementary Fig. 10.5). However, previous models[30] illustrated that increasing roughness may cause an increase of the *effective* specific energy of adhesion $\gamma_{eff}$ (Fig. 7f) ($\gamma_{eff}$ is the work of detachment of the cell from the rough substrate). With Gentile[23], the effective energy-density of adhesion is related, in turn, to the lateral forces that act on cells during adhesion and that can eventually enhance cell clustering and coalescence (Fig. 7g). Thus, in the end, residual roughness on the substrate is responsible for the increased adhesion and increased clustering of cells on soft materials (Supplementary Fig. 10.6). In simulation in which rough PDMS surface of lateral size $l = 10\,\mu m$ and initial value of roughness Ra $\sim 20\,nm$ was discretized into $n = 100$ elements, we found that for a cell-surface binding energy density of $\gamma = 10^{-5}J/m^2$, the value of roughness shifts to about Ra $\sim 30\,nm$ ($E_{substrate} = 1MPa$). As a result, the intensity of lateral forces on cells moves from $\sim 3nN$ to about $\sim 5nN$ ($E_{cell} = 10kPa$, membrane thickness $= 10nm$) – that for a sufficiently high density of cells may mark the transition from non-clustered to clustered systems. Details of the model are reported in a separate Supplementary Note 10.

In this study, we employed established techniques for collecting and culturing primary hippocampal neurons[73]. Although this approach might yield a mix of neurons and glial cells, previous research indicates that the presence of glial cells is significantly less critical than that of neurons[39,74,75]. To demonstrate that glial cells constitute a minor fraction of the total cell population on the substrate surfaces, we conducted further experiments (Supplementary Note 11). After 15 days from seeding, we captured and analyzed fluorescence images of hippocampal cells cultured on traditional flat surfaces. We stained cell nuclei with DAPI, which indiscriminately marks both neuronal and glial cells, alongside selective staining with anti-NeuN antibodies that target neuron nuclei specifically. This dual-staining approach allowed us to assess the colocalization of all cultured cell nuclei with those of neurons. We quantified colocalization through metrics, such as the Pearson correlation coefficient ($Pc = 0.66$) and Manders colocalization coefficients[76,77] ($M_1 = 0.965$) and ($M_2 = 0.74$), which reveal the extent of signal overlap between the two channels (Supplementary Fig. 11.1). Our findings, with $M_1$ indicating a high overlap between neuronal cells and the total cell population and $M_2$ showing that approximately 75% of the cells identified by DAPI staining are neurons, align with literature reporting that neurons account for 65% to 80% of cells in the mouse hippocampus[78,79]. Thus, our analysis confirms that neurons predominantly occupy the cultured surfaces. Significantly, these observations were made after 15 days in vitro, a period considerably longer than that used in this study for network characterization (i.e., up to 4 days), highlighting the predominantly neuronal makeup of the cultures even as glial cells are known to proliferate over time. This evidence strongly supports our conclusion that the data and insights derived from this study predominantly reflect neuronal activity. Additionally, much of our research relies on functional multi-calcium imaging (fMCI) experiments, which inherently focus on neuronal activity and are unaffected by glial or other non-neuronal cells. The outcomes of these fMCI experiments, demonstrating a correlation between neuronal activity and substrate compliance, corroborate our observations of neuron adhesion and clustering on PDMS. They reinforce our finding that substrate elasticity inversely affects neuronal performance within the low MPa range.

This research demonstrates that adjusting the stiffness of materials could be a valuable strategy for creating more effective substrates for biological applications. The computational model presented here offers crucial insights for selecting the appropriate physical parameters for specific uses. For example, substrate properties like stiffness and roughness can be customized for having, in one case, substrates with a higher propensity in favoring homogeneously spread and firmly adhered cells (low connectivity) in the case of biological systems not requiring the processing of large amount of information. In another case, such parameters can be optimized for the realization of more complex and highly interconnected biological systems, as for a brain-like tissue, or even for providing the suitable architecture for artificial organs. It's important to note that the studied parameters are not directly tied to the substrates' chemical composition or structure. Yet, they significantly influence neuronal cell organization and clustering. This suggests that a material's stiffness and geometric properties might be as crucial as its surface chemistry in directing cell behavior. Therefore, the design and creation of materials for tissue engineering, regenerative medicine, and experimental models for neuro-degeneration studies should consider a combination of mechanical, chemical, and geometric characteristics. These factors collectively impact essential qualities of biomaterials like bio-compatibility, biodegradability, and overall performance.

## Methods

### Fabrication of soft PDMS surfaces

Soft polydimethylsiloxane (PDMS) substrates were produced using replica molding techniques. A template of empty discs was obtained by micro-machining polymethyl methacrylate (PMMA). Each disc had a diameter and thickness of $1cm$ and $2mm$, respectively. Then, a solution of liquid PDMS and curing/solidifying agent (SYLGARD™ 184 Silicone Elastomer Kit, Dow Corning) was poured into a baker, gently stirred, and put under vacuum for approximately 30 minutes to enable degasification and remove unwanted bubbles. Then, the solution was poured into the template, placed in an oven and cured at 80℃ for 60 minutes. After cooling, samples were carefully peeled from the template. All samples were inspected by naked eye and optical microscopy for detecting possible defects and, once passed the quality check, stored in a Petri dish for future use. The pre-polymer ($pp$) and the curing agent ($ca$) were mixed using a ratio $r = pp : ca$ varying between 7 : 1 and 18 : 1. The larger the amount of pre-polymer in solution, the smaller the stiffness of the resulting substrates.

### Mechanical characterization of PDMS surfaces

Mechanical characterization of the PDMS was carried out by both tensile and nano-indentation tests. While the tensile test has the advantage of providing highly reliable results, independent from the shape and the dimensions of the specimen, nano-indentation test is non- or semi-destructive, and does not require specific specimen preparation.

Tensile tests were carried out following the methods reported in ref. 80 and recapitulated in a separate Supplementary Note 1. Specimens, in the form of strips with a cross-section of $5mm \times 2mm$, were tested by the universal testing machine MTS model Criterion 42, equipped with the load cell LSB.102. The load was applied by manual vice-action grips. All samples were tested in displacement control mode by setting the speed of the moving cross-head to $2mm/$ min. Images of samples under load were acquired by a Prosilica ATV-GT2450 camera. Then, the VIC-2D software (Correlated Solutions) was used to evaluate the *in-plane* displacement components from the acquired images. We used digital image correlation (DIC) algorithms to determine the strain field as a gradient of the displacement field in a selected ROI.

Nano-indentation tests were performed following the methods reported in[80,81]. PDMS samples were placed under the tip of a nano-indentation machine (Anthon-Parr instrumented hardness station). The penetration depth and the load were then registered for each sample. The Young's modulus was then determined from the slope of the unloading curve at the early stage of the process (Supplementary Note 1).

### Measuring topography of PDMS surfaces

We acquired topographical images of soft PDMS surfaces using a Trib-ometer MFT-5000 with an integrated 3D profilometer from Rtec. The profilometer, equipped with 50 × and 100 × objectives, operated in non-contact/confocal mode to generate 3D interferometric profiles of the surface

under white light illumination as illustrated, as for an example, in ref. [82]. Surface topography images were acquired by performing vertical z-axis scans of the samples, using a scan length of XX. During each scan, the step-size and frame-rate of acquisition were set as $2.5 nm$ and $7 frames/s$, respectively. Resulting images had a size of 1280 (width) × 960 (height) pixels, and a pixel size of $\sim 78 nm$. For each sample, the total imaged area was then of $\sim 100 \times \sim 75$ microns. For each PDMS-curing agent ratio and sample stiffness, we performed at least 5 different measurements. The mean ($Ra$) and root mean square ($Rrms$) values of roughness were then evaluated as $Ra = \int_{S_A} |z(x, y)|/S_A$ and $Rrms = (\int_{S_A} z^2(x, y)/S_A)^{1/2}$, respectively, where $z$ is the sample height, $x$ and $y$ are the spatial coordinates in the plane of the measurement, and $S_A$ is the sampling area.

### Determining the fractal dimension of PDMS surfaces
The height profiles of PDMS surfaces were processed using the methods reported in[38] and recapitulated in a separate Supplementary Note 12.

### Measuring surface contact angle
The wetting characteristics of the soft PDMS surfaces were determined using an automatic contact angle meter (KSV CAM 101, KSV Instrumetns LTD, Helsinki, Finland). One small drop ($\sim 5 \mu l$) of deionized water was gently positioned on the surface, and the contact angle between the solid, liquid, and air phases measured at room temperature $5 s$ after drop casting. For each substrate, the contact angle $\vartheta$ was averaged over 4 repeated measurements. The energy of adhesion $\gamma$ per unit area at the PDMS/water interface was then determined as $\gamma = \gamma_{la}(1 + \cos \vartheta)$, where $\gamma_{la} \sim 72.8 mJ/m^2$ is the surface tension between air and water.

### Primary neuronal cultures on PDMS substrates
Soft PDMS substrates were individually placed in 12-multi-well plates (Corning Incorporated) and sterilized under UV irradiation for $12 h$. Poly-D-lysine (PDL) (Sigma-Aldrich, Milan, Italy) was diluted in sterile $H_2O$ to a final concentration of $1 \mu g/ml$ and used to cover the substrates prior neuronal cells culture. Substrates were left in the PDL solution overnight in a cell culture incubator ($37 °C, 5\% CO_2, 5\%$ humidity). No treatment with oxygen plasma was—remarkably—employed on PDMS surfaces before coating with PDL. Notwithstanding, PDMS sample surfaces were moderately hydrophilic with values of contact angle less than $80°$ for all considered substrate preparations, as evidenced by measurements reported in Fig. 2h. Neuronal cells were extracted from C57B/L6 mouse embryos brains at day 18 (E18) as described in previous works[39,43]. All procedures were carried out in accordance with the guidelines established by the European Communities Council (Directive of November 24th, 1986) and approved by the National Council on Health and Animal Care (authorization ID 227, prot. 4127, 25th March 2008). Pregnant females were deeply anesthetized with $CO_2$ and decapitated. Embryos were then removed, brains were extracted and placed in cold Hank's Balanced Salts solution (HBSS). Upon removal of the meninges, the hippocampus was dissected, incubated with 0.125% trypsin for 15 min at $37 °C$ and dissociated. Neurons were plated on the PDMS substrates in complete cell-culture medium, supplemented with 10% fetal bovine serum (FBS, Invitrogen), 5% penicillin G (100 U/ml) and streptomycin sulfate (100 mg/ml) (Invitrogen). Then, neurons were incubated at $37 °C$ in a humidified $5\% CO_2/air$ atmosphere with a density of $10^5$ cells/ml. Neurons were incubated for different periods, 24, 48, 72, 96 hours in vitro, to evaluate the effect of time on cell clustering. Neurons were plated with the same density on PDL-coated rigid polystyrene substrates serving as a control. Cells were sub-confluent throughout the duration of the experiment.

### Neuronal cells staining
After incubation, the cell culture medium was removed and the cells were washed twice in PBS. Then, they were fixed with 4% PFA (paraformaldehyde) and incubated for 30 min at room temperature (RT). The cells were washed twice PBS and made permeable with 0.05% triton (Invitrogen) for 5 min at RT. The nuclei of the cells were then stained with $100 \mu l$ DAPI

(40, 6-Diamidino-2-phenylindole, Sigma Aldrich) solution for 10 min at $4 °C$ at dark. The actin filaments of the cells were labelled using $100 \mu l$ of green fluorescent staining phalloidin conjugate (cytopainted, from Abcam, Italy), incubated with the cells for 15 min at $4 °C$ at dark. At the end of staining, the DAPI and phalloidin conjugate solutions were removed and each sample washed with PBS.

### Neuronal cells imaging
At the end of the incubation period and after staining, cells adhering on the substrates were imaged using a Nikon ECLIPSE Ti fluorescent microscope using the methods reported in a separate Supplementary Note 13.

### Image analysis
Fluorescent images of cells were imported in Matlab R2020b and converted from RBG to gray scale format. The $k$-means algorithm was applied to segment the images and select the sole region occupied by cells. The k-means clustering algorithm partitions the originating images into k different segments[42,83]. The information content of the image was associated to one of the segments and all the other segments were disregarded as background. The segment containing the information was shifted to black pixels (binary 1), while the background was associated to white pixels (binary 0). $k$ depends upon the particular problem at study and, for the present configuration, it was set as $k = 8$. After segmentation, a grid with a mesh size of $8 \times 8$ pixels was applied to the images to determine the average intensity color of each mesh and associate this value to the probability of being a cell. A threshold (80% of the maximum color intensity) was applied to define the presence of a cell. The region was shrunk to a single pixel and associated with a node, corresponding to the center of the cell.

### Neuronal cells wiring
Once that the neuronal cell centers were determined, they were connected using a mixed distance and density rule described elsewhere[58,62] and reported in a separate Supplementary Note 14.

### Cell-networks analysis
Upon wiring, the information about the connections between cells was stored in the adjacency matrix. The adjacency matrix $A_{ij}$ is a matrix with a number of rows and columns equal to the number of cells measured for each sample in a region of interest. The elements of $A_{ij}$ are such that if $A_{ij} = 1(0)$ cell $i$ and cell $j$ are connected (disconnected). The adjacency matrix was then used to calculate for each network the small world (SW) coefficient defined as:

$$sw = \frac{cc}{cc_{random}} \bigg/ \frac{cpl}{cpl_{random}} \qquad (3)$$

where $cc$ is the mean clustering coefficient of the network averaged over all the individual clustering coefficients calculated for each node as:

$$cc_i = 2E_i/n(n - 1) \qquad (4)$$

In Eq. (4), $E_i$ is the number of the existing connections about a node $i$, $n$ is the degree of the node, and $n(n - 1)/2$ is the maximum number of connections that can be established around the node[57]. $Cpl$ is the characteristic path length defined as the averaged shortest path length ($spl$) among all the combinations of nodes in the network, taken two at the time[57]. $cc_{random}$ and $cpl_{random}$ are the clustering coefficient and characteristic path length of an Erdős–Rényi random graph with the same size and degree of the network under study. Thus, the small-world coefficient of a graph is larger than one if that network has higher clustering and shorter paths than an equivalent random graph with the same size.

### Simulating information flows in neuronal cell networks
Networks of neuronal cells built after fluorescence images of cells were stimulated with an external disturbance to understand how a different

network topology on soft PDMS surfaces affects cell-signaling. The model of propagation is described in a separate Supplementary Note 15.

## Decoding information contained in the spike trains

The train of 0/1 bits recorded for each neuron over time can be decoded using the method of information theory described in refs. 65–67. The method involves partitioning the signal in finite sequences of time and calculating how many times each sequence appears in the originating pattern of 0/1 values. Results of the calculation are a distribution of frequencies as a function of a specific state, $P(s)$. The Shannon information entropy associated to $P(s)$ is then $H(S) = -\sum_s P(s) log_2 P(s)$, where $S$ stands for stimulus[65]. $H$ is a quantitative estimate of the information carried by $S$. The information transported in the network is thus $I = H(S_1) - H(S_2)$, where $S_1$ ($S_2$) is a random (periodic) signal of time[84].

## Functional multicalcium imaging

Methods relative to the fMCI experiments are reported in a separate Supplementary Note 16.

## Statistics and reproducibility

Results in the main article and supplementary material are reported as mean ± standard deviation. In determining the number and the topological characteristics of neurons on soft surfaces (Figs. 3 and 4) we performed 3 sample repeats. Overall, for each considered value of material stiffness and cell incubation time, we processed and analyzed 30 to 50 images. A minimum sample size of 30 images was chosen to increase the confidence interval of the population data set. Analysis of variance (ANOVA) was used to examine whether the differences between SW means measured on substrates with varying elasticity are statistically significant (Fig. 5). In performing the test, the null hypothesis is that the means between pairs of samples are equal. Everywhere in the text and the figures the difference between two subsets of data is considered statistically significant if the ANOVA test gives a significant level less than 0.05.

## Data availability

All data underlying charts and diagrams reported in Figs. 2–6 are deposited in the public data repository OSF under the name "Soft Neurons" (https://doi.org/10.17605/OSF.IO/WV34G). Data supporting this study are available within the paper and the Supplementary Information. All other data are available from the authors upon reasonable request.

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

## Acknowledgements
This study was partially funded by the Italian Association for Cancer Research (AIRC) under the grant number AIRC IG 2021 ID 25656. The Authors thank Andrea Contestabile for the support provided during the fMCI experiments.

## Author contributions

G.M. analysed data and determined the small-world coefficient of neuronal networks. L.B. performed mechanical characterization of soft PDMS surfaces. N.P. fabricated surfaces. M.L.C. supervised the experimental part of the work and analysed data. M.N. prepared neuronal cultures for the fMCI experiments. N.M. prepared neuronal cell cultures for direct investigation of networks by fluorescence microscopy. E.B. examined leakage of PDMS by EDAX analysis. G.B. imaged sample morphology by SEM and performed EDAX measurements of PDMS. F.D.A. supervised SEM and EDAX analysis of samples. L.C. supervised the fMCI experiments. D.D.M. performed the fMCI experiments and helped to write the MS. F.G. conceived the project, devised and implemented the mathematical model of cell-adhesion and clustering on soft surfaces, wrote the paper and supplementary information and directed the investigations. All authors discussed the results and commented on the manuscript.

## Competing interests
The authors declare no competing interests.
