## [Peer Review File · Communications Biology]

Reviewers' comments:

Reviewer #1 (Remarks to the Author):

The work submitted by Gentile et al. reports on the combined effect of Young's modulus, small-worldness and roughness on neuronal cell adhesion, both experimentally and numerically. The investigation is sound, extremely well written and very innovative in this field, therefore I am in favour of its publication in *Communications Biology* after addressing the following remarks:

- Provide additional information about the features of the probe employed for nanoindentation (stiffness, material, shape) and explain why you employed an Oliver-Pharr model instead of a Hertzian one.
- Was oxygen plasma employed on PDMS before PDL coating?
- Provide SEM characterization of the samples and elaborate further on the cell/biomaterial interaction as well as on the morphological changes of neuronal cells.

Reviewer #2 (Remarks to the Author):

The manuscript by Marinaro et al has significant methodological issues listed below, and authors conclusions are not supported by the results.

1. Authors used different ratios of pre-polymer base and curing agent of a commercial formulation of PDMS, Sylgard 184, to create substrates of different stiffness. However, authors did not take into account that altering base/curing agent ratio may alter chemical and polymer network properties of PDMS that may affect cells in many different ways. Besides siloxane units, the prepolymer base comprises a vinylated and methylated surface-modified silica filler component and vinyl-terminated and branched siloxane components. The curing agent also contains linear and cyclosiloxanes, the modified silica filler, and platinum to catalyze crosslinking of PDMS network. The crosslinked networks that result from changing ratios of base and curing agent may be different not only in mechanical properties, but also in their water and other molecule absorption, release of filler and non-crosslinked PDMS over time, interaction between polymer surface and poly-D-lysine, and many other properties. Many of these can affect cellular attachment and viability. It is therefore difficult to make any conclusions based on authors' data in regards to effect of substrate stiffness on cultured neurons.

2. The authors claim that data shown in Figure 3 indicates a linear relationship between stiffness and the number of cells. However, in panels 3(b), 3(c), and 3(f) the highest number of cells corresponded to an intermediate 1 MPa elasticity, while in panel 3(d) highest number of cells corresponded to 2 MPa elasticity, with both higher and lower elasticity substrates resulting in fewer cells. The data does not seem to support authors hypothesis. Instead, it appears that there is an optimal PDMS mixing ratio that results in best cell viability for unknown reason.

3. Data in Fig. 3 shows a substantial decrease in the number of cells at 72 hours followed by an increase at 96 hours. Authors do not explain this. From methods, it appears that cultures were performed in serum-based medium, so there may be division of glial cells at later time points overlapping death of neurons. This would complicate interpretation, as no neuronal identification (neuronal markers) was performed.

4. Neuronal networks were derived from positions of cell bodies (which may not even be neurons) and

clustering. But neurons connect via axon and dendrite growth. These neurites may grow differently on different substrates – not clear that same distance-based rules can/should be applied for different conditions. No attempt was made to trace neuronal connectivity via either axon tracing or physiological methods.

5. Calcium imaging method was not described; section called functional multi calcium imaging in Supplementary Information describes dissection method instead.

Reviewer #3 (Remarks to the Author):

The manuscript presented by Marinaro et al. has reported the impact of mechanical properties such as stiffness etc. on neuronal cell proliferation and its network formation. They have studied the topology of their PDMS substrates and conducted a detailed theoretical analysis based on the neuron distribution fluorescent images. Their work has practiced the method of small-world network (published in 2022) and, in the meantime, proposed a mechanical model to further explain the mechanism of how soft materials influencing cells. The overall quality of this work is good but a few issues need to be addressed, which are listed below:

Major:

1. Introduction, paragraph 4, page 3, line 69-71. The authors have cited some previous studies and discussed the contradiction in the role of stiffness. I notice that those cited studies were mainly working on relatively soft materials with modulus in the range of 2kPa to 65kPa (e.g. Lo et al. ref #46: 61-300 kdyn/cm², which is about 6.1kPa-30kPa; Ng. et al. ref #47: 2-65kPa; Kamimura et al. ref #45: 5-55kPa, etc.) while the authors studied on the PDMS materials with modulus in the range of 0.55-2.65MPa, which is relatively stiffer. Therefore, the authors' current work may not be sufficient enough to resolve this contradiction. It is highly suggested that the authors may want to reconsider their statements and revise properly, starting from a general background but focusing on a detailed aspect that the current results can handle.

In addition, the contradiction that emphasized by the authors could be caused by multiple factors. Those cited previous works have used different types of cells such as fibroblasts (Lo et al.), mammary epithelial cells (Ng et al.), kidney epithelial cells (Kamimura et al.), etc. from different tissues, and these cells are highly differentiated, which may require a unique microenvironment for proliferation. Probably none of these studies could provide a general answer at this moment due to cell type, substrate material property, even 2D or 3D culture environment, etc., thereby, contradictory could exist.

2. "Results" section, "Small-world characteristics of neuronal cell networks on PDMS surfaces" subsection, page 9, line 250. The authors concluded that "the values of SW steadily decrease with E for all the considered probabilities p - supporting the findings of this results section", which is questionable. According to the supporting information 5 provided by the authors, supporting information figures 5.1 and 5.3 have fittings showing SW not decreasing with the increase of E.

3. Equation (2), page 12. Is the equation (2) missing a square to dy/dx ? (should it be $(dy/dx)^2$?) Please double check the data if this mistake exists and influence the analysis.

4. "Discussion and conclusions" section, "Perspectives in biomedical engineering" subsection, page 14. The authors stated, "Thus, even materials, which are usually not appropriate for biological applications, but have other features that can be helpful in certain processes, can become suitable as biomedical material by tuning their mechanical properties". This statement may need additional consideration. Under the scope of this work, indeed, all studied parameters are not strictly dependent on the materials' chemical compositions. However, under the scope of biological applications, it is inefficient to only have proper mechanical properties. Other factors such as biocompatibility should also be considered. The authors should avoid exaggeration in a scientific article.

5. Figure 2d. Figure 2d is slightly confusing. Is there an overall linear tilting on the surface? Since the y-axis in the figure (2d right) indicates the height or dimensions in z-direction and different colored lines correspond to four sampling dashed lines from the left figure. However, this linear tilting is not shown in the figure on the left. Instead, there is a blue-yellow-blue repeated pattern on the background in figure 2d left, which may be another type of background unevenness. The authors may want to revise or supplement additional figures or content accordingly to avoid this potential confusion. In addition, is the colored scale bar for figure 2e also applicable to figure 2d? Since figure 2d (left) does not have a legend or scale bar to explain the relation between the color and height.

Minor:

1. Abstract: The first two sentences in the abstract may need to be reconsidered and revised. The authors have stated a very broad background with a very general question in the first sentence and then followed a statement starting with "To address the role of elasticity as a regulator of cell adhesion and clustering, ...", which sounds like the authors were going to resolve this general question mentioned above. However, the purpose of this work is actually focusing on the impact of physical properties on neuronal cell networks. Therefore, to clarify the goal better, it is suggested that the authors should revise their statement to, "To address the role of elasticity as a regulator in neuronal cell adhesion and clustering, ..." or similar.

2. "Results" section, "Measuring PDMS surface characteristics" subsection, page 5, line 143. The term "different-from-zero value of roughness" was not mentioned in the later content. Considering the length and abundance of the content, it is suggested that the authors could provide an explanation right after bringing in this concept or rephrase the viewpoint properly.

3. Figure 3h. It could be confusing and improper to use the infinity symbol to represent the control group. Please use "control", "ctrl", or other proper symbols to represent the control group.

4. Scale bar issues. Scale bars are missing and should be provided in figure 3a, figure 4 a&b, figure 5b, figure 6 a&b, and in all figures in supporting information 3.

Reviewers' comments:

Reviewer #1 (Remarks to the Author):

The work submitted by Gentile et al. reports on the combined effect of Young's modulus, small-worldness and roughness on neuronal cell adhesion, both experimentally and numerically. The investigation is sound, extremely well written and very innovative in this field, therefore I am in favour of its publication in *Communications Biology* after addressing the following remarks:

We really thank the Reviewer for the positive comments on the MS.

Q1.1 Provide additional information about the features of the probe employed for nanoindentation (stiffness, material, shape) and explain why you employed an Oliver-Pharr model instead of a Hertzian one.

We appreciate the observation of the Reviewer.

To comply with his observation, we report in the Supporting Information 1 additional information about the indenter used in the tests. Specifically, the nanoindenter is a Berkovich tip, i.e. a three-sided pyramid which is geometrically self-similar. The indenter tip material is diamond, with Young's modulus of 1141 GPa and a value of Poisson's ratio equal to 0.07. To the best of our knowledge, there are not methods based on the Hertzian model useful to retrieve the mechanical properties of a material from an indentation test. The consolidated methods of measurement of hardness and elastic modulus by instrumented indentation rely on the Oliver-Pharr model or the tangent model. Further to this point, the software included with the indentation station is limited to using either the Oliver-Pharr method or the tangent method for data analysis purposes. The main assumption of the Oliver-Pharr model, is that the Young's modulus is proportional to the *slope* of the unloading stress-strain curve. In this phase, the mechanical behavior of a material is elastic - an assumption, in turn, of Hertzian theory. Consequently, the Oliver-Pharr model is in harmony with the Hertzian model and effectively integrates it.

Q1.2 Was oxygen plasma employed on PDMS before PDL coating?

Oxygen plasma was not used on PDMS before coating with PDL. Notwithstanding, PDMS sample surfaces were moderately hydrophilic with values of contact angle less than 80° for all considered substrate preparations. Using oxygen plasma on sample surfaces can increase even further wettability, promoting cell adhesion and proliferation, possibly amplifying the topological effects of cells observed in this research. This observation has been conveniently inserted in the text.

Q1.3 Provide SEM characterization of the samples and elaborate further on the cell/biomaterial interaction as well as on the morphological changes of neuronal cells.

In agreement with the observation of the Reviewer, we have performed SEM analysis of PDMS sample surfaces. The analysis was performed using a dual beam Helios Nanolab 600 (Thermo Fisher) scanning electron microscope. Samples were sputtered with 10 nm of gold before imaging and glued to a standard focused ion beam/SEM stub with silver glue. The images were carried out with the sample tilted at 45° in immersion mode, and the back-scattered electrons were collected with a current of 0.20

nA and a 3 kV acceleration of the electronic beam. **Figure R1.1**, below, illustrates the morphology of a sample surface - for a PDMS:curing agent ratio of 14 and relative value of elastic modulus of 1 MPa. The morphology of PDMS measured by SEM is consistent with the values of surface roughness determined by laser interferometry of about 20 nm.

The same SEM analysis was performed on PDMS sample surfaces with values of elasticity of 0.55, 1.88, 2.65 MPa (**Figure R1.2**). In all considered examples, surface profile is never perfectly flat. Instead, it always exhibits some spatial variability. Surface texture is consistent with morphology measurement performed by quantitative interferometry analysis. Results of this characterization campaign have conveniently included in a separate Supporting Information file.

A thorough analysis of cell/biomaterial interaction is already included in the paper, in a separate supporting information. The mathematical model of adhesion and clustering that we have developed, explains with good accuracy results of the research: that in the low mega-Pascal range neuronal cells form well organized structures on soft surfaces.

As regarding the morphological changes of cells. While this aspect is of interest and deserves to be investigated, however an accurate analysis of cell morphology as a function of substrate stiffness necessitates high magnification fluorescence microscopy images of cells, and a custom experimental campaign. A more sophisticated evolution of the research that will be performed over time, will be aimed at dissecting the relationship between cell morphology and cell topology on soft surfaces.

r-1:14 / E-1. MPa

Figure R1.1. PDMS surface morphology examined by scanning electron microscopy for a sample preparation corresponding to a value of elasticity of 1 MPa.

r-1:7 / E-2.65 MPa

r-1:10 / E-1.88 MPa

r-1:14 / E-1. MPa

r-1:18 / E-0.55 MPa

Figure R1.2. PDMS surface morphology examined by scanning electron microscopy for a sample preparation corresponding to a values of elasticity spanning the 0.55-2.65 MPa range.

Reviewer #2 (Remarks to the Author):

The manuscript by Marinaro et al has significant methodological issues listed below, and authors conclusions are not supported by the results.

Q2.1. Authors used different ratios of pre-polymer base and curing agent of a commercial formulation of PDMS, Sylgard 184, to create substrates of different stiffness. However, authors did not take into account that altering base/curing agent ratio may alter chemical and polymer network properties of PDMS that may affect cells in many different ways. Besides siloxane units, the prepolymer base comprises a vinylated and methylated surface-modified silica filler component and vinyl-terminated and branched siloxane components. The curing agent also contains linear and cyclosiloxanes, the modified silica filler, and platinum to catalyze crosslinking of PDMS network. The crosslinked networks that result from changing ratios of base and curing agent may be different not only in mechanical properties, but also in their water and other molecule absorption, release of filler and non-crosslinked PDMS over time, interaction between polymer surface and poly-D-lysine, and many other properties. Many of these can affect cellular attachment and viability. It is therefore difficult to make any conclusions based on authors' data in regards to effect of substrate stiffness on cultured neurons.

We agree with the observation of the Reviewer.

While we have examined the mechanical and morphological characteristics of PDMS substrates, we have not sufficiently-well investigated their chemical properties, with the exception of the sole angle of contact. The contact angle reflects the wettability of a material - that in turn depends on its surface energy of adhesion γ through the celebrated Young - Dupré equation $\gamma = \gamma_{la}(1 + \cos\theta)$. Since the energy of adhesion determines the extent of adhesion of cells on a surface (1), the findings of our study that surface energy density is approximately constant for different substrate preparations (with a variation below 6%) suggests that the chemical properties of samples are not relevant to the conclusions of the research. - And that the effects of stiffness and surface chemistry can be conveniently decoupled.

However, to comply with the observation of the Reviewer, and dissect even further whether the interaction between neuronal cells and the PDMS material may depend on other factors other than stiffness - we have characterized sample surfaces using Raman spectroscopy and Energy Dispersive X-Ray Analysis (EDAX).

Raman spectra of PDMS samples were obtained using a Renishaw InVia Microscope with a 1024 CCD detector, an excitation wavelength of 633 nm, and 50× objective lens. The laser power was maintained at a constant value of 300 mW and the integration time was set to 50 s throughout all measurements. We acquired Raman spectra of samples at different proportions ρ of the SYLGARD 184 polymeric base to the curing agent. Values of ρ were varied from 7: 1 to 18: 1 - the elasticity of resulting samples changed accordingly from ~2.65 to ~0.55 MPa. To assure statistical significance, for each sample we performed 7x7 measurements over a square region of $70 \mu m \times 70 \mu m$. After acquisition, Raman spectra were base line corrected to remove background. Then, we performed min-max normalization and cosmic rays removal (2). Remarkably, resulting spectra (Figure R2.1) exhibit peaks typical of polydimethylsiloxane (3, 4), and specifically at: 491.8, 615.6, 707.6, 786.0, 847.7, 1242.8, 1393.3 cm^{-1} .

That correspond to the following vibrational modes: Si–O–Si stretching (491.8 cm^{-1}); Si–C asymmetrical (615.6 cm^{-1}), symmetrical (707.6 cm^{-1}) and combined (786.0 cm^{-1}) stretching; CH₃ asymmetrical stretching (847.7 cm^{-1}) and asymmetrical bending (1393.3 cm^{-1}); SiCH₃ symmetrical bending (1242.8 cm^{-1}).

Figure R2.1. Raman spectrum and average Raman spectrum of PDMS surfaces for different material elasticity values.

Figure R2.1 illustrates Raman spectra of PDMS samples for different sample-preparations. For each value of surface elasticity, Raman spectra of the same sample are reported in the same diagram. In the diagrams, the black, thick-line graph is the average spectrum. Visual, *qualitative* inspection of graphs suggests that there are minimal differences between samples. To measure *quantitatively* how the Raman signature of PDMS samples varies as a function of sample preparation, we report in **Figure R2.2** the mean-peak intensity as a function of surface elasticity, evaluated for values frequency typical of PDMS, i.e. 491.8 , 615.6 , 707.6 , 786.0 , 847.7 , 1242.8 , 1393.3 cm^{-1} . Peak-intensity variations are moderate for all considered frequencies, with an overall variance percentage comprised between 2% and 11%

(Figure R2.2h). This suggests that the chemical surface-structure of samples is subject to minimal-important changes as substrate elasticity varies from 0.55 to 2.65 MPa - and the proportion ρ between the SYLGARD 184 polymeric base and the curing agent varies from 7:1 to 18:1.

Figure R2.2. Raman intensity measured in correspondence of frequencies characteristic of PDMS as a function material elasticity (a-g). Numerical values of mean Raman intensities measured at different central frequencies, for different values of PDMS elasticity (h).

We further analyzed Raman spectra using principal components analysis (PCA). PCA is a multivariate technique of analysis that can reduce the dimensionality of a dataset by finding the variables that contribute to its variance (2), i.e. the principal components (PCs).

Raman spectra relative to all PDMS sample preparations were grouped in a single dataset and processed. After the analysis, we found that the first three components - PC1, PC2 and PC3 - account for more than the 99% of the information content of the original data, thus we used PC1, PC2 and PC3 to represent the spectra in the following of the study. Scatter plots of PC1 vs PC2 (Figure R2.3a), of PC2 vs PC3 (Figure R2.3b) and of PC1 vs PC3 (Figure R2.3c) indicate that data points are homogeneously

distributed in the space of the principal components – without a clear clustering of points into separate groups. This in turn indicates that there are not statistically significant differences among Raman spectra associated to different sample preparations. This is also evidenced by the decision graph (Figure R2.3d) relative to the data points in the PC space. The decision graph is a plot of the *density* against the *minimal distance to other points with greater density* of points in a metric space (5). In a similar graph, points that stand out from the other with high values of *density* and relatively high values of *distance* mark the cluster centers (5). Lack of outliers in the diagram in Figure R2.3d hints that there is not a clear distinction between sample groups – and that different sample preparations cannot be segregated by Raman analysis.

Figure R2.3. PCA analysis of PDMS Raman spectra. Scatter plot of PC1 vs PC2 (a), PC2 vs PC3 (b) and PC1 vs PC3 (c) relative to the Raman spectra of PDMS for different values of material elasticity varying from 0.55 to 2.65 MPa. Decision graph relative to the principal components associated to Raman spectra of PDMS surfaces for different values of material elasticity varying from 0.55 to 2.65 MPa (d).

We then performed an additional Raman analysis on samples coated with poly-d-lysine (PD) to examine whether different PDMS sample preparations can affect PDMS-PD interaction. Since Raman peaks characteristic of PD fall in the 800-1800 $1/\text{cm}$ range (6, 7), the measurement range was set to this

interval. Raman spectra of PD-coated PDMS - reported in Figure R2.4a for the sole $E = 2$ MPa substrate - illustrate that the peaks typical of the components of PD are vanishingly small, perhaps sheltered by PDMS. With the remarkable exception of the peak at 1276 $1/\text{cm}$, a hallmark of the amide III band of α -helix. Diagrams in Figure R2.4b show how the mean Raman spectrum of PD-coated-samples changes as a function of PDMS elasticity. The Raman intensity measured at 1276 $1/\text{cm}$ is reported in the diagram of Figure R2.4c as a function of material elasticity. Values of Raman intensity vary from 0.06 for $E=0.55$ MPa, to 0.064 for $E=2.65$ MPa. In the considered elasticity interval, the maximum variation of intensity is of $e\sim 36\%$, more relevant than the variations relative to the sole PDMS, but still less relevant than the variation in Young's modulus, that is of $\sim 380\%$.

Figure R2.4. Raman spectra and average Raman spectrum of PD-coated PDMS for value of Young's modulus $E=2$ MPa (a). Average values of Raman spectrum of PD-coated PDMS for different values of PDMS elasticity varying from 0.55 to 2.65 MPa (b). Trend of Raman intensity measured at 1276 $1/\text{cm}$ as a function of the material Young's modulus (c).

Principal Components Analysis of Raman spectra indicates that the 1st, 2nd and 3rd principal components of Raman spectra do not seemingly cluster into groups (Figure R2.5a-c), as illustrated by the decision graph relative to these data points Figure R2.5d. Thus Raman analysis and PCA processing of data

suggest that the interaction between poly-d-lysine and PDMS is only marginally influenced by PDMS preparation.

Figure R2.5. PCA analysis of PD-coated PDMS Raman spectra. Scatter plot of PC1 vs PC2 (a), PC2 vs PC3 (b) and PC1 vs PC3 (c) relative to the Raman spectra of PDMS coated with PD for different values of material elasticity, varying from E-0.55 to E-2.65 MPa. Decision graph relative to the principal components associated to Raman spectra of PDMS surfaces coated with PD, for different values of material elasticity varying from 0.55 to 2.65 MPa (d).

To the end to characterize with maximum precision the chemical composition of PDMS samples, we further performed Energy dispersive X-ray analysis (EDAX) of samples. We performed EDAX on PDMS samples with a polymeric-base:curing-agent ratio varying between 7:1 and 18:1 – and Young's modulus spanning the 0.55-2.65 MPa range. We used for the analysis a dual beam Helios Nanolab 600 (Thermo Fisher) scanning electron microscope. Samples were mounted on a standard stub and analyzed with a probe current of 0.20 nA and 5kV acceleration of the electronic beam. Results reported in Table R2.1 illustrate that the composition of carbon (C), oxygen (O) and silicon (O) varies only marginally for different substrate preparations.

Sample #	ρ	E (MPa)	C (weight %)	O (weight %)	Si (weight %)
1	7	2.65	52.35	44.99	2.66
2	8	2.5	51.62	44.9	3.39
3	9	2	51.59	44.76	3.65
40	10	1.88	48.45	44.86	6.73
5	14	1	51.56	44.76	3.34
6	18	0.55	50.02	44.87	5.11

Table R2.1. Elemental analysis of PDMS samples.

Overall, results presented in this section and conveniently reported in a separate supporting information, suggest that the polymeric-base:curing-agent ratio ρ influences significantly the Young's modulus of PDMS samples, and affects marginally the chemical composition and structure of samples.

1. B Majhy, P Priyadarshini and A. K. Sen. Effect of surface energy and roughness on cell adhesion and growth – facile surface modification for enhanced cell culture. *RSC Adv.*, 2021, 11, 15467-15476

2. D Di Mascolo, A Coclite, F Gentile and M Francardi. Quantitative micro-Raman analysis of micro-particles in drug delivery. *Nanoscale Advances*, 2019, 1, 1541-1552.

3. Angel S. Cruz-Felix, Agustin Santiago-Alvarado, Josimar Marquez-García, Jorge Gonzalez-García. PDMS samples characterization with variations of synthesis parameters for tunable optics applications. *Heliyon* 5: e03064, 2019.

4. A Zahid, B Dai, R Hong and D Zhang. Optical properties study of silicone polymer PDMS substrate surfaces modified by plasma treatment. *Mater. Res. Express* 4: 105301. 2017.

5. Alex Rodriguez and Alessandro Laio, Clustering by fast search and find of density peaks, *Science* 322(6191): 1492-1496, 2014.

6. Danielle Carrier, Michel Pezolet. Raman spectroscopic study of the interaction of poly-L-lysine with dipalmitoylphosphatidylglycerol bilayers. *Biophysical Journal* 46(4): 497–506, 1984.

7. Julian F. A. Perlitz, Lukas Gentner, Phillipp A. B. Braeuer and Stefan Will. Measurement of Secondary Structure Changes in Poly-L-lysine and Lysozyme during Acoustically Levitated Single Droplet Drying Experiments by In Situ Raman Spectroscopy. *Sensors* 22: 1111, 2022.

Q2.2. The authors claim that data shown in Figure 3 indicates a linear relationship between stiffness and the number of cells. However, in panels 3(b), 3(c), and 3(f) the highest number of cells corresponded to an intermediate 1 MPa elasticity, while in panel 3(d) highest number of cells corresponded to 2 MPa elasticity, with both higher and lower elasticity substrates resulting in fewer cells. The data does not seem to support authors hypothesis. Instead, it appears that there is an optimal PDMS mixing ratio that results in best cell viability for unknown reason.

We thank the Reviewer for the observation.

Diagrams in Figure 3d-f illustrate how the number of neurons (N) varies as a function of surface-elasticity (E) in the 0.5-2.65 MPa range, for different incubation times. While the N-E relationship is not perfectly linear, however in the considered interval the number of neurons *overall* decreases with the Young's modulus. For all the time points of the analysis, N is always higher on softer ($E \sim 0.55 \text{ MPa}$) than on harder ($E \sim 2.65 \text{ MPa}$) surfaces. In the 0.55-2.65 MPa range, the association between N and E is moderate to strong – and never weak – as evidenced by values of r-squared of $r^2 \sim 0.64$ for $t=24 \text{ h}$, $r^2 \sim 0.51$ for $t=48 \text{ h}$, $r^2 \sim 0.69$ for $t=72 \text{ h}$, $r^2 \sim 0.54$ for $t=96 \text{ h}$. Further to this end: for each considered time, if one of the data-points were excluded from the analysis, still the E(N) function would remain decreasing.

Thus, we decided to maintain the conclusion that the number of neurons decreases with surface elasticity – especially considering that in the text we have specified that the N-E relationship is only *approximated* by a linear function.

In the revised version of the manuscript we have correctly reported the values of r-squared determined from the linear model fit of experimental data, for each of the considered time points.

Q2.3. Data in Fig. 3 shows a substantial decrease in the number of cells at 72 hours followed by an increase at 96 hours. Authors do not explain this. From methods, it appears that cultures were performed in serum-based medium, so there may be division of glial cells at later time points overlapping death of neurons. This would complicate interpretation, as no neuronal identification (neuronal markers) was performed.

We appreciate the observation of the Reviewer.

The observed variation in N moving from 72 to 96 h for all considered surface elasticities might depend on statistical fluctuations in the experiment and the measurements. Notice though that in the 24h-96h interval, the maximum calculated rate of change ($\partial N / \partial t$) is of approximately 4 cells/day, i.e. negligible and consistent with the notion that neurons do not proliferate.

Also notably, the same variation from a lower (at 72 h) to a higher (at 96 h) number of cells - is observed in the control, i.e. substrates with theoretically infinite elasticity. Thus, the oscillation of N measured for all substrate types may simply reflect an initial not-accurate, unbalanced, distribution of cells during cell seeding.

Q2.4. Neuronal networks were derived from positions of cell bodies (which may not even be neurons) and clustering. But neurons connect via axon and dendrite growth. These neurites may grow differently on different substrates – not clear that same distance-based rules can/should be applied for different conditions. No attempt was made to trace neuronal connectivity via either axon tracing or physiological methods.

We appreciate the observation of the Reviewer. In this study - the topological characteristics of neuronal networks have been determined starting from the positions of the nuclei on the surface, that have been then elaborated through convenient wiring models. However, in agreement with the observation of the Reviewer, this represents just an *estimate* of how cells connect. A similar model based on cell-distance may explain less neuronal cell connectivity and more cell-condensation and clustering.

Figure R2.6. Image analysis and processing of fluorescent images of cells, aimed at the determining the most faithful representation of neuronal cell networks.

To generate more reliable neuronal cell networks and provide a consolidated reference against which results of the work can be verified, we performed additional analysis. In place of examining cell-nuclei, we analyzed neuronal branching from green fluorescent images of cells, in which actin filaments were labelled using green fluorescent staining phalloidin conjugate – as conveniently reported in the methods of the paper. Since actin filaments are expressed in subcellular structures such as growth cones or dendritic spines, they can be used to dissect neurite outgrowth or synapse physiology.

For different sample characteristics, we examined green-fluorescent images of neuronal cells lining the PDMS surface (Figure R2.6-a). Images were then gray-scale converted and enhanced using a bi-lateral filter (Figure R2.6-b). After correction, images were skeleton-transformed, this enabled to reduce foreground regions in the originating image, preserving the extent and connectivity of the original region while throwing away most of the original foreground pixels (Figure R2.6-c). After removing smaller disconnected objects, the morphological graph of the cells was determined - giving the morphological branch points and endpoints of the image (Figure R2.6-d). The very good overlap of the axons in the originating fluorescent image of cells, with the topological-neural network (Figure R2.6-e) is an evidence of the correctness of the method that we have developed to find the graph-analogue of biological neurons. Then, the topological characteristics of the network g depicted in Figure R2.6-d and e - were compared to those determined for an Erdos-Renyi random graph with the same size of g (Figure R2.6-f) to determine the small-world coefficient of the cells. Thus, this image analysis procedure and algorithm enable to generate a faithful analogue of real neuronal graphs (Figure R2.7).

Figure R2.7. Neuronal-network analogue (b) of real neurons (a) – determined by the neuronal branching/image analysis algorithm described in Figure R2.6.

For the cells considered in this example (i.e. the control) we found that $SW=1.004$, practically the same as the value $SW=1$ estimated in the study through simple neurons-wiring. The same analysis was performed for cells cultured for 48 h on PDMS surfaces with values of elasticity $E_1=0.55$ MPa, $E_2=1$ MPa, $E_3=1.88$ MPa and $E_4=2.65$ MPa (Figure R2.8).

Results illustrate that the small-world coefficient of neuronal cell graphs decreases linearly with the Young's modulus (Figure R2.9) – in line with the results already presented in the study obtained by connecting nodes using a mixed distance and density rule wiring model.

Figure R2.8. Image analysis and processing performed on cells sitting on surfaces with different values of elasticity and polymeric-base:curing-agent mixing ratio.

Specifically, the linear law that correlates the small-world coefficient (sw) to the Young's modulus of PDMS surfaces (E) obtained using the Waxman model on neurons - imaged 48h from culture - is $SW = 2.42 - 0.59 E$. In contrast, the relationship found by more-accurate image analysis and processing of fluorescent images of neuronal cells - is $SW = 1.98 - 0.62 E$.

The very close resemblance between linear model fits obtained using different procedures indicates that results of this research study based on a cell-cell-distance and cell-density wiring model - are accurate.

This analysis has been conveniently inserted in a separate supporting information file.

Figure R2.9. Linear relationship between the small world-coefficient of neuronal networks determined from fluorescent images of cells, and the elasticity of PDMS cell-culture substrates.

Q2.5. Calcium imaging method was not described; section called functional multi calcium imaging in Supplementary Information describes dissection method instead.

In agreement with the observation of the Reviewer, we have revised the calcium imaging methods in the supporting information, to include details on sample preparations and image acquisition, as follows:

“ Functional multi-calcium imaging. PDMS surfaces with varying compliance, hosting neuronal cells in networks, were incubated in 1 ml of dye solution (Fluo4, Life Technologies/Thermo Fisher, at a 1:2000 dilution) at room temperature for 20 min in dark conditions. Samples were then washed once with PBS and immediately imaged over time with a fluorescent upright microscope (Leica DM6000, Wetzlar, Germany), using the following parameters: Exposure time = 30 ms; Frame rate = 300 ms. The overall images were acquired with the LAS AF software (Leica Microsystems Srl).”

Reviewer #3 (Remarks to the Author):

The manuscript presented by Marinaro et al. has reported the impact of mechanical properties such as stiffness etc. on neuronal cell proliferation and its network formation. They have studied the topology of their PDMS substrates and conducted a detailed theoretical analysis based on the neuron distribution fluorescent images. Their work has practiced the method of small-world network (published in 2022) and, in the meantime, proposed a mechanical model to further explain the mechanism of how soft materials influencing cells. The overall quality of this work is good but a few issues need to be addressed, which are listed below:

We thank the Reviewer for the positive comments on the MS.

Major:

Q3.1. Introduction, paragraph 4, page 3, line 69-71. The authors have cited some previous studies and discussed the contradiction in the role of stiffness. I notice that those cited studies were mainly working on relatively soft materials with modulus in the range of 2kPa to 65kPa (e.g. Lo et al. ref #46: 61-300 kdyn/cm², which is about 6.1kPa-30kPa; Ng. et al. ref #47: 2-65kPa; Kamimura et al. ref #45: 5-55kPa, etc.) while the authors studied on the PDMS materials with modulus in the range of 0.55-2.65MPa, which is relatively stiffer. Therefore, the authors' current work may not be sufficient enough to resolve this contradiction. It is highly suggested that the authors may want to reconsider their statements and revise properly, starting from a general background but focusing on a detailed aspect that the current results can handle.

In addition, the contradiction that emphasized by the authors could be caused by multiple factors. Those cited previous works have used different types of cells such as fibroblasts (Lo et al.), mammary epithelial cells (Ng et al.), kidney epithelial cells (Kamimura et al.), etc. from different tissues, and these cells are highly differentiated, which may require a unique microenvironment for proliferation. Probably none of these studies could provide a general answer at this moment due to cell type, substrate material property, even 2D or 3D culture environment, etc., thereby, contradictory could exist.

In agreement with the observation of the Reviewer, we have revised this part of the introduction.

Firstly, we have correctly indicated that results of the existing body of literature on the role of stiffness on cell adhesion are *seemingly* contradictory. And that the different behavior of cells (in terms of adhesion, proliferation, migration, differentiation) observed by different groups on soft or hard surfaces, may be ascribed to several factors other than stiffness – including chemical structure of the material, *cell type*, the experimental or environmental conditions of the measurements.

Then, following suggestion from the Reviewer, we have correctly commented on the fact that the cited literature focused on materials with elasticity in the 2kPa to 65kPa range. A highly cited review and a reference (1) for those working in the field of biomaterials and regenerative medicine maintains that – in this elasticity range – *while it is generally true that increasing substrate stiffness correlates with increasing cell differentiation, there are many exceptions, and the stiffness optimum for differentiation*

and other behaviors varies significantly from cell to cell. Further to this end, in another seminal work (2), it is reported that epithelial cells on soft gels (with $E \sim 1$ kPa) show diffuse and dynamic adhesion complexes; in contrast, stiff gels (with $E \sim 100$ kPa) show cells with stable focal adhesions. In the same work, it is recalled that, in the low kPa range, soft deformable substrates enhance neurite branching. Thus, *again*, the lack of consensus on whether material stiffness enhances or undermines cell activity is - most likely - credited to the heterogeneity of conditions under which the great many of these studies have been performed.

Few studies have explored cell-surface interactions for relatively harder materials - with values of elasticity in the low MPa range.

In reference (3) it is illustrated that osteogenic differentiation and mineralisation by embryonic stem cells is enhanced on substrates higher Young's modulus (>2.3 MPa) - when compared to softer substrates with E in the $0.04 - 1.9$ MPa range. In another study (4), it is shown that polyurethane films with high values of Young's modulus (higher than approximately 4 MPa) enhance the adhesive capacity of NIH 3T3 fibroblasts and Wharton's jelly mesenchymal stem cells. The article presented in reference (5) investigated how the properties of substrates influence the fate of stem cells. Researchers cultivated individual human epidermal stem cells on surfaces of polydimethylsiloxane (PDMS) and polyacrylamide (PAAm) hydrogels, varying in stiffness from 0.1 kPa to 2.3 MPa, with collagen coating attached covalently. They observed that the stiffness of PDMS did not affect cell spreading and differentiation. Conversely, on low-stiffness PAAm (0.5 kPa), cells failed to form stable focal adhesions and differentiated. Human mesenchymal stem cell differentiation was similarly independent of PDMS stiffness but was influenced by PAAm's elastic modulus. Analysis of dextran penetration revealed that less stiff PAAm substrates were more porous; suggesting that surface roughness can play a role during adhesion.

Our work fits into this long-standing series of previous studies. The aim of this research is elucidating how substrate elasticity – in the low 0.55 - 2.65 MPa range – influences neuronal growth, networking, and activity.

We have conveniently incorporated these comments and revised part of the introduction as follows:

"Notice though that these previous works have used different types of cells such as fibroblasts (Lo et al.), mammary epithelial cells (Ng et al.), kidney epithelial cells (Kamimura et al.), from different tissues, and these cells are highly differentiated, which may require a unique microenvironment for proliferation. Thus, the different behavior of cells (in terms of adhesion, proliferation, migration, differentiation) observed by different groups on soft or hard surfaces, may be ascribed to several factors other than stiffness – including chemical structure of the material, cell type, the experimental or environmental conditions of the measurements.

In addition, notice that the cited, existing body of literature focused on relatively soft materials with elasticity in the 2kPa to 65kPa range. A highly cited review and a reference (1) for those working in the field of biomaterials and regenerative medicine maintains that – in this elasticity range – while it is generally true that increasing substrate stiffness correlates with increasing cell differentiation, there are

many exceptions, and the stiffness optimum for differentiation and other behaviors varies significantly from cell to cell. Further to this end, in another seminal work (2), it is reported that epithelial cells on soft gels (with $E \sim 1$ kPa) show diffuse and dynamic adhesion complexes; in contrast, stiff gels (with $E \sim 100$ kPa) show cells with stable focal adhesions. In the same work, it is recalled that, in the low kPa range, soft deformable substrates enhance neurite branching. Thus, again, the lack of consensus on whether material stiffness enhances or undermines cell activity is, most likely, credited to the heterogeneity of conditions under which the great many of these studies have been performed.

Few studies have explored cell-surface interactions for relatively harder materials - with values of elasticity in the low MPa range. In reference (3) it is illustrated that osteogenic differentiation and mineralisation by embryonic stem cells is enhanced on substrates higher Young's modulus (>2.3 MPa) - when compared to softer substrates with E in the $0.04 - 1.9$ MPa range. In another study (4), it is shown that polyurethane films with high values of Young's modulus (higher than approximately 4 MPa) enhance the adhesive capacity of NIH 3T3 fibroblasts and Wharton's jelly mesenchymal stem cells. The article presented in reference (5) investigated how the properties of substrates influence the fate of stem cells. Researchers cultivated individual human epidermal stem cells on surfaces of polydimethylsiloxane (PDMS) and polyacrylamide (PAAm) hydrogels, varying in stiffness from 0.1 kPa to 2.3 MPa, with collagen coating attached covalently. They observed that the stiffness of PDMS did not affect cell spreading and differentiation. Conversely, on low-stiffness PAAm (0.5 kPa), cells failed to form stable focal adhesions and differentiated. Human mesenchymal stem cell differentiation was similarly independent of PDMS stiffness but was influenced by PAAm's elastic modulus. Analysis of dextran penetration revealed that less stiff PAAm substrates were more porous; suggesting that surface roughness can play a role during adhesion.

Our work fits into this long-standing series of previous studies. The aim of this research is elucidating how substrate elasticity – in the low 0.55 - 2.65 MPa range – influences neuronal growth, networking, and activity. To do this, we fabricated soft polydimethylsiloxane (PDMS) surfaces by replica molding techniques (Figure 1a-b) – that were used as a substrate for neuronal cell culture and grow (Figure 1c). Neuronal cells were then examined at 24, 48, 72, 96 h from culture by fluorescence microscopy and also by functional multi calcium imaging (fMCI) (Figure 1d). Fluorescence images of cells were processed using image analysis, network analysis, and information theory algorithms Figure 1e-f. Results of the analysis indicate that – in the considered elasticity range – adhesion and connectivity of neuronal cells are optimized for small values of elasticity of the substrate.”

Moreover, to comply even further with the observation of the Reviewer, we have conveniently changed the title of the MS into:

“In the low MPa range, material compliance accelerates the development of topologically-efficient networks of neuronal cells”.

1. Rebecca G. Wells. The Role of Matrix Stiffness in Regulating Cell Behavior. HEPATOLOGY, Vol. 47, No. 4, 2008.

2. Dennis E Discher, Paul Janmey, Yu-Li Wang. Tissue cells feel and respond to the stiffness of their substrate. *Science* 310(5751):1139-43. 2005.
3. Evans, N. D., Minelli, C., Gentleman, E., LaPointe, V., Patankar, S. N., Kallivretaki, M., et al. (2009). Substrate stiffness affects early differentiation events in embryonic stem cells. *Eur. Cell Mater.* 18, 1–13.
4. Gaëtan Lutzweiler, Julien Barthes, Nihal Engin Vrana, Michel Rawiso, Benoît Louis, Josselin Mayingi, Albane Carre, Wiebke Drenckhan, and Pierre Schaaf. Adjustment of Cell Adhesion on Polyurethane Structures via Control of the Hard/Soft Segment Ratio. *Macromol. Mater. Eng.* 2020, 305, 2000093.
5. Britta Trappmann, Julien E. Gautrot, John T. Connelly, Daniel G. T. Strange, Yuan Li, Michelle L. Oyen, Martien A. Cohen Stuart, Heike Boehm, Bojun Li, Viola Vogel, Joachim P. Spatz, Fiona M. Watt and Wilhelm T. S. Huck. Extracellular-matrix tethering regulates stem-cell fate. *Nature Materials* 11, 642–649 (2012).

Q3.2. “Results” section, “Small-world characteristics of neuronal cell networks on PDMS surfaces” subsection, page 9, line 250. The authors concluded that “the values of SW steadily decrease with E for all the considered probabilities p - supporting the findings of this results section”, which is questionable. According to the supporting information 5 provided by the authors, supporting information figures 5.1 and 5.3 have fittings showing SW not decreasing with the increase of E.

We appreciate the observation of the Reviewer.

In the original version of the paper, we have inaccurately claimed that - the values of SW decrease with E for all the considered probabilities p - . In agreement with the observation of the Reviewer, we have revised the sentence as:

“[...] the values of SW decrease with E for all the considered probabilities p –with the exception of $p=0.96$ at 24 h, and $p=0.86$ and $p=0.88$ at 72h from culture. Thus results of this extended simulation campaign mostly support the findings of this section”.

Also notice that in a separate analysis, we have demonstrated that results of our wiring model and network building - are consistent with direct reconstruction of neuronal cell graphs by neuronal branching analysis from green fluorescent images. Results of this additional analysis – produced in response to an observation of the 2nd Reviewer and conveniently reported in a separate supporting information file - indicate that the main findings of this research study based on a cell-cell distance and cell-density wiring model of cells – are *accurate*.

Specifically, that the small-world characteristics of neuronal networks are hindered by surface stiffness in the low MPa range.

Q3.3. Equation (2), page 12. Is the equation (2) missing a square to dy/dx ? (should it be $(dy/dx)^2$?) Please double check the data if this mistake exists and influence the analysis.

We thank the Reviewer for the observation.

It was just a typo that has been corrected in the revised version of the MS and of the SI. The correct form of the radius of curvature is, as indicated by the Reviewer:

$$R(x) = \left(1 + \left(\frac{dy}{dx}\right)^2\right)^{3/2} / \left|\frac{d^2y}{dx^2}\right|$$

In the originating Mathematica file used for the simulations, we have used the right form of the equation. Thus, results of the work are not in any way affected by this typo.

Q3.4. “Discussion and conclusions” section, “Perspectives in biomedical engineering” subsection, page 14. The authors stated, “Thus, even materials, which are usually not appropriate for biological applications, but have other features that can be helpful in certain processes, can become suitable as biomedical material by tuning their mechanical properties”. This statement may need additional consideration. Under the scope of this work, indeed, all studied parameters are not strictly dependent on the materials' chemical compositions. However, under the scope of biological applications, it is inefficient to only have proper mechanical properties. Other factors such as biocompatibility should also be considered. The authors should avoid exaggeration in a scientific article.

In agreement with the observation of the Reviewer, we have revised the section “perspectives in biomedical engineering” as reported below – diminishing the claims on the importance of the sole mechanical properties as a regulator of cell behavior:

“Perspectives in biomedical engineering.

This research demonstrates that adjusting the stiffness of materials could be a valuable strategy for creating more effective substrates for biological applications. The computational model presented here offers crucial insights for selecting the appropriate physical parameters for specific uses. For example, substrate properties like stiffness and roughness can be customized for having, in one case, substrates with a higher propensity in favoring homogeneously spread and firmly adhered cells (low connectivity) in the case of biological systems not requiring the processing of large amount of information. In another case, such parameters can be optimized for the realization of more complex and highly interconnected biological systems, as for a brain-like tissue, or even for providing the suitable architecture for artificial organs.

It's important to note that the studied parameters are not directly tied to the substrates' chemical composition or structure. Yet, they significantly influence neuronal cell organization and clustering. This suggests that a material's stiffness and geometric properties might be as crucial as its surface chemistry in directing cell behavior. Therefore, the design and creation of materials for tissue engineering, regenerative medicine, and experimental models for neuro-degeneration studies should consider a combination of mechanical, chemical, and geometric characteristics. These factors collectively impact essential qualities of biomaterials like biocompatibility, biodegradability, and overall performance.”

Q3.5. Figure 2d. Figure 2d is slightly confusing. Is there an overall linear tilting on the surface? Since the y-axis in the figure (2d right) indicates the height or dimensions in z-direction and different colored lines correspond to four sampling dashed lines from the left figure. However, this linear tilting is not shown in the figure on the left. Instead, there is a blue-yellow-blue repeated pattern on the background in figure 2d left, which may be another type of background unevenness. The authors may want to revise or supplement additional figures or content accordingly to avoid this potential confusion. In addition, is the colored scale bar for figure 2e also applicable to figure 2d? Since figure 2d (left) does not have a legend or scale bar to explain the relation between the color and height.

We appreciate the observation of the Reviewer.

We have improved the appearance of Figure 3 to avoid confusion. Firstly, we have base-line corrected the density-plot of sample profile using a two-dimensional complete 6-th degree polynomial function of the form: $P_6(x, y) = \sum_{k=0}^6 a_k x^i x^j, i + j \leq k$, as shown in Figure R3.1. We have then correctly included a legend in the inset (d) of the figure – specifically relative to the 2d density-plot of sample surface. Then, we have correctly removed the ticks from the y-axis of figure 3.d left, added a scale bar, and used one single color for the four profile-lines shown in figure 3.d right.

Please, notice that removal of the background (i.e. polynomial correction) does not affect in any way results of the paper and the values of roughness reported in the following of the work – since they have been themselves calculated with respect to a base-line.

Figure R3.1. Base-line correction of sample surface morphology.

Minor:

Q3.6. Abstract: The first two sentences in the abstract may need to be reconsidered and revised. The authors have stated a very broad background with a very general question in the first sentence and then followed a statement starting with “To address the role of elasticity as a regulator of cell adhesion and clustering, ...”, which sounds like the authors were going to resolve this general question mentioned above. However, the purpose of this work is actually focusing on the impact of physical properties on neuronal cell networks. Therefore, to clarify the goal better, it is suggested that the authors should revise their statement to, “To address the role of elasticity as a regulator in neuronal cell adhesion and clustering, ...” or similar.

Following suggestion from the Reviewer, we have revised the initial sentences of the abstract -

“The question of whether material’s stiffness enhances cell adhesion and clustering is still open to debate. Results from the literature are *seemingly* contradictory, with some reports illustrating that adhesion increases with surface stiffness and others suggesting that the performance of a system of cells is curbed by high values of elasticity. Elasticity may be particularly important in relation to the development and shaping of the brain, in that cells of the brain, neurons, incessantly interact with the extracellular matrix – an intricate network of macromolecules and minerals that provide *structural* and functional support. To address the role of elasticity as a regulator in neuronal cell adhesion and clustering, [...].”

Q3.7. “Results” section, “Measuring PDMS surface characteristics” subsection, page 5, line 143. The term “different-from-zero value of roughness” was not mentioned in the later content. Considering the length and abundance of the content, it is suggested that the authors could provide an explanation right after bringing in this concept or rephrase the viewpoint properly.

In agreement with the observation of the Reviewer, we have rephrased the sentence as follows:

“The characteristics of sample surfaces of being rough, with values of roughness in the 10-20 nm range, can be accountable for the peculiar behavior of cells – that on soft PDMS substrates cluster into defined groups more markedly compared to harder substrates. The possible mechanisms and hypothesis underlying a similar behavior are explained in the following of the paper and in a separate supporting information.”

Q3.8. Figure 3h. It could be confusing and improper to use the infinity symbol to represent the control group. Please use "control", "ctrl", or other proper symbols to represent the control group.

In agreement with the observation of the Reviewer, we have used in ctrl in place of infinity, to label the control group.

Q3.9. Scale bar issues. Scale bars are missing and should be provided in figure 3a, figure 4 a&b, figure 5b, figure 6 a&b, and in all figures in supporting information 3.

Following indication of the Reviewer, scale bars have been added where appropriate, in figures of the main article and the supporting information.

Reviewers' comments:

Reviewer #1 (Remarks to the Author):

The Hertzian model is widely used for relatively soft biomaterials nanoindentation experiments (see e.g. <https://iopscience.iop.org/article/10.1088/2053-1591/ab79ce> and <https://doi.org/10.1016/j.jmbbm.2022.105329>).

The current title should be revised as its current form is not adequate and sounds like a main text sentence.

The rest of my remarks have been addressed.

Reviewer #2 (Remarks to the Author):

Revision of the manuscript by Marinaro et al does not address most of my concerns, particularly points 2, 3, and 4 raised in the original review.

1. Authors carried out new experiments to characterize chemical composition of Sylgard 184 made with different base/curing agent ratios. However, authors did not make an attempt to quantify degree of cross-linking and potential leakage of uncured PDMS into culture medium. This is a well-known problem that is probably exacerbated by changing the amount of curing agent used, as authors did in this manuscript. This in itself could explain toxicity that is apparent in Figure 3 at highest and lowest extremes of base/curing agent ratios – which may be the more dominant effect compared to mechanical effects that authors are interested in. Two references describing PDMS leaching are listed at the bottom of this review. On a positive note, authors do provide Raman spectra showing the presence of bonds one would expect from PDMS with little apparent variability between different PDMS preparations. EDAX data is not as positive: authors show that Si percentage changes by more than a factor of 2 between different PDMS preparations (Table R2.1 in the Rebuttal). This confirms my concern that changes in PDMS chemical composition may be the reason for authors' results, rather than any mechanical effect.

2. I am still concerned about authors claiming a linear relationship based on data that is inverse U-shaped (Figure 3, panels 'c' and 'f' in particular). What are the sample numbers and p-values for linear correlation coefficients authors are reporting? This is important because the inverse U-shape of the data suggests toxicity at extreme low and high base/curing agent ratios.

3. In the rebuttal, authors state that a decrease in cell numbers at 72 hrs and subsequent increase in cell numbers at 96 hrs is due unbalanced distribution of cells during cell seeding. This suggests a very high experimental or measurement variability – this experiment does not appear to have been conducted with scientific rigor. Authors did not address the concern about neuronal viability or glial cell proliferation: these processes can be concurrent (i.e. neurons could be dying, while glia could be dividing), resulting in unpredictable changes in overall cell number. In general, authors do not seem to be aware that primary hippocampal cultures contain proliferative (especially in 10% FBS) glial cells in addition to neurons. This is a major weakness of the manuscript.

4. Authors presented a new image-based method of assessing formation of networks in culture. Unfortunately, the method is not neuron specific. Phalloidin staining was used to label actin filaments – but this method can also label glial processes. There is no evidence that networks authors have derived from culture images are actually neuronal due to lack of neuron-specific staining markers.

5. Authors added description of the calcium imaging method.

References.

Regehr KJ, Domenech M, Koepsel JT, Carver KC, Ellison-Zelski SJ, Murphy WL, Schuler LA, Alarid ET, Beebe DJ. Biological implications of polydimethylsiloxane-based microfluidic cell culture. *Lab Chip*. 2009 Aug 7;9(15):2132-9. doi: 10.1039/b903043c. Epub 2009 Jun 4. PMID: 19606288; PMCID:

PMC2792742.

Sarah-Sophia D. Carter, Abdul-Raouf Atif, Sandeep Kadekar, Ingela Lanekoff, Håkan Engqvist, Oommen P. Varghese, Maria Tenje, Gemma Mestres, PDMS leaching and its implications for on-chip studies focusing on bone regeneration applications, *Organs-on-a-Chip*, Volume 2, 2020, 100004, ISSN 2666-1020, <https://doi.org/10.1016/j.ooc.2020.100004>.

Reviewer #3 (Remarks to the Author):

The authors have sufficiently resolved my concerns. I have no further comments regarding this manuscript. And I would like to thank the authors for their efforts in the revision.

Reviewer #1 (Remarks to the Author):

The Hertzian model is widely used for relatively soft biomaterials nanoindentation experiments (see e.g. <https://iopscience.iop.org/article/10.1088/2053-1591/ab79ce> and <https://doi.org/10.1016/j.jmbbm.2022.105329>).

In acknowledgment of the observation of the Reviewer, we have conveniently commented in the Supporting Information that the Hertzian model, together with the Oliver & Pharr's, can be used to explain and interpret AFM nanoindentation experiments on biological samples.

The current title should be revised as its current form is not adequate and sounds like a main text sentence.

Following the suggestion from the Reviewer, we have correctly changed the title of the MS into “The role of elasticity on adhesion and clustering of neuronal cells on soft PDMS surfaces in the low MPa range”. This reflects more accurately the contents of the works and is less a statement, compared to the original title “In the low MPa range, material compliance accelerates the development of topologically efficient networks of neuronal cells”.

The rest of my remarks have been addressed.

We appreciate this observation and we are pleased to have properly addressed all the concerns previously raised by the Reviewer.

Reviewer #2 (Remarks to the Author):

Revision of the manuscript by Marinaro et al does not address most of my concerns, particularly points 2, 3, and 4 raised in the original review.

1. Authors carried out new experiments to characterize chemical composition of Sylgard 184 made with different base/curing agent ratios. However, authors did not make an attempt to quantify degree of cross-linking and potential leakage of uncured PDMS into culture medium. This is a well-known problem that is probably exacerbated by changing the amount of curing agent used, as authors did in this manuscript. This in itself could explain toxicity that is apparent in Figure 3 at highest and lowest extremes of base/curing agent ratios – which may be the more dominant effect compared to mechanical effects that authors are interested in. Two references describing PDMS leaching are listed at the bottom of this review. On a positive note, authors do provide Raman spectra showing the presence of bonds one would expect from PDMS with little apparent variability between different PDMS preparations. EDAX data is not as positive: authors show that Si percentage changes by more than a factor of 2 between different PDMS preparations (Table R2.1 in the Rebuttal). This confirms my concern that changes in PDMS chemical composition may be the reason for authors' results, rather than any mechanical effect.

Following the suggestions of the Reviewer, we have performed an additional test campaign aimed at characterizing leakage of PDMS into DI water at different times and temperatures. To do so, we used both Raman spectroscopy and energy dispersive X-ray spectroscopy (EDX).

RAMAN

As regarding Raman analysis.

PDMS substrates, molded into cylindrical shapes with a diameter of 2 cm and a height of 0.5 cm using varying ratios of the liquid PDMS to curing agent ratio ($r:1$, $r=7$, 10, 13, 15, 18), were submerged in 200 μl of water to assess their stability. The testing was conducted at two different temperatures: 37°C , to mimic cell culture conditions, and 60°C , to accelerate the leaching process. Additionally, the substrates were incubated for a time varying from 24 h (short incubation period) to 108 h (long incubation period). To analyze any product that may have been released following the long-standing exposition of PDMS to water, a drop from each solution was placed in duplicate on a CaF_2 slide and on a CaF_2 slide coated with a sputtered gold layer, and then allowed to dry (**Figure R2.1**). The samples were subsequently examined using a Renishaw inVia Raman microscope (Renishaw, Turin, Italy) at ambient temperature, employing a $50\times$ objective of a Leica microscope (Leica Micro Systems, Wetzlar, Germany). Raman spectra were collected by exciting the samples with a 630.0 nm laser line, at a laser power of 2.5 mW and an integration time of 10 to 20 s.

Figure R2.1. *Experimental scheme.* PDMS discs were exposed to DI water for a time varying from 24 h to 108 h, setting two different values of the leaching temperature, i.e. $T_1 = 37\text{ }^\circ\text{C}$ and $T_2 = 60\text{ }^\circ\text{C}$, and for different values of the liquid base:curing agent ratio, r . The solution resulting from the prolonged contact was collected, deposited on a substrate and left to evaporate. The residue was then examined by Raman and EDX spectroscopy (a). Optical images of PDMS traces released into DI water, for different values of the PDMS-water exposure time and for a fixed leaching process temperature $T = 60\text{ }^\circ\text{C}$ (b-d).

Analysis of PDMS residue using simple CaF_2 substrates fails to detect clear signatures of sample leakage, probably due to the low amplification effects of CaF_2 , the weak signal associated with PDMS byproducts, and with unwanted fluorescence interference (Figure R2.2 - R2.3). Raman spectra reported in Figure R2.2 are relative to samples obtained under a fixed temperature of $T=37\text{ }^\circ\text{C}$ after 48 and 108 h of exposition. Raman spectra reported in Figure R2.3 are relative to samples obtained under a fixed temperature of $T = 60\text{ }^\circ\text{C}$ after 48 and 108 h of exposition. In both cases, Raman spectra exhibit no evident peaks attributable to PDMS components in the remnant of solution after exposure, sample collection, and evaporation.

Figure R2.2. Raman signature of PDMS traces, for different values of the leaching time, and of the liquid base:curing agent ratio r . For a fixed value of the leaching temperature $T_1 = 37^\circ\text{C}$.

Figure R2.3. Raman signature of PDMS traces, for different values of the leaching time, and of the liquid base:curing agent ratio r . For a fixed value of the leaching temperature $T_2 = 60^\circ\text{C}$.

To amplify even more the signal and generate significant Raman spectra, we used SERS (Surface Enhanced Raman Spectroscopy) substrates, obtained by sputtering a discontinuous layer of gold nanoparticles upon an originating CaF_2 flat surface (**Figure R2.4**).

Figure R2.4. Optical and SEM image of a CaF_2 substrate patterned with gold nanoparticles.

Raman spectra of samples measured on similar SERS surfaces deliver more information about the PDMS components released over time and under different working temperatures, compared to conventional CaF_2 substrates. Results of SERS measurements are reported in **Figure R2.5** for leaching times of 48 and 108 h, for a PDMS-base/curing agent ratios (r) of $r = 7, 10, 13,$ and 18 ; and for a fixed external temperature $T = 37^\circ\text{C}$. After acquisition, Raman spectra were background corrected and normalized to the maximum value in the measurement range. Results of the analysis illustrate that there are peaks characteristic of PDMS traces in solution, specifically:

- Peaks at 784 1/cm and 830 1/cm . While weak, they show an increasing trend from the $r=7$ to $r=18$ samples. This trend can be attributed to the Si-C stretch, as indicated in reference (1).
- Peak at 1051 1/cm : This peak exhibits a decreasing trend from the $r=7$ to $r=18$ samples. It is likely attributed to the C-C stretch, as documented in reference (1), possibly due to the methyl residue from the curing agent.

In **Figure R2.5**, for the central frequency 784 1/cm (Si-C stretch) we report the mean-peak intensity as a function of the PDMS to curing agent ratio, r : a quantitative measure of how the Raman signature of PDMS samples varies as a function of sample preparation, for both the 48 h and 108 h time steps. After 48 h of leakage, peak variations across different r 's are moderate, with the intensity varying of about the 8% for r moving from $r=7$ to $r=13$, and of about 4% for r moving from $r=13$ to $r=18$. After 108 h of leakage, the variation of intensity relative to the 784 1/cm frequency remains moderate, with the signal varying of about the 5% for r moving from $r=7$ to $r=18$.

Figure R2.5. Raman signature of PDMS traces, for different values of r and of the leaching time (48 h, 108 h), for a fixed leaching temperature $T_1 = 37$ °C. In the figure, we also report the percentage variation of the Raman peak (ΔI) relative to the frequency $f = 784 \text{ cm}^{-1}$, determined as a function of r for both of the considered times of leaching, 48 h and 108 h.

Figure R2.6 illustrates the same analysis reported in **Figure R2.5**, except that the temperature of the leakage process is set to $T=60$ °C. For this configuration, the Raman signal measured at 784 1/cm (typical of Si-C stretch) varies of less than the 3% in the 7-18 PDMS:curing agent interval for a 48 h test time, and of a vanishingly small 1% for a 48 h test time.

Results of the PDMS leakage tests performed by Raman analysis suggest that sample preparation (i.e. the ratio r) affects more relevantly the mechanical characteristics of samples (i.e. the Young's modulus), while it influences only moderately the leakage of PDMS in a liquid solution.

This analysis enhances our confidence that the results of the work and the observed, peculiar behavior of neuronal cells are directly related to the mechanical properties of PDMS samples, rather than being influenced by other factors such as leakage.

Figure R2.6. Raman signature of PDMS traces, for different values of r and of the leaching time (48 h, 108 h), for a fixed leaching temperature $T_2 = 60\text{ }^\circ\text{C}$. In the figure, we also report the percentage variation of the Raman peak (ΔI) relative to the frequency $f = 784\text{ cm}^{-1}$, determined as a function of r for both of the considered times of leaching, 48 h and 108 h.

EDX

Since the Raman analysis that we have performed is sensitive more to the molecular composition of samples and less to its elements, we carried out an additional sample test by using energy dispersive X-ray spectroscopy (EDX). Differently from Raman spectroscopy, this technique is sensitive to the elements constituting the samples. Moreover, EDX can be used to estimate the relative abundance of elements in a sample. To perform the tests, a 20 μl drop of DI-water in contact with PDMS under different conditions was deposited on a clean standard SEM pin stub and analyzed by FESEM ULTRA-PLUS (Zeiss) (Milan, Italy) with the SE2 detector.

We performed the analysis on samples exposed at different leakage temperatures (37 $^\circ\text{C}$, 60 $^\circ\text{C}$), for different incubation times (24 h, 48 h, 108 h), and with different initial PDMS:curing agent ratios (r : 7, 10, 13, 15). The results of the analysis are reported in Figures R2.7 to R2.13.

Figures R2.7 and R2.8 illustrate results of an EDX analysis performed on samples obtained from the interaction between DI water and PDMS, following 48 h (R2.7) and 108 h (R2.8) of exposure at a steady incubation temperature of $T=37^{\circ}\text{C}$, in relation to the liquid base-to-curing agent ratio, r .

Similarly, Figures R2.9 to R2.11 illustrate results of an EDX analysis performed on samples obtained from the interaction between DI water and PDMS, following 24 (R2.9), 48 (R2.10) and 108 h (R2.11) of exposure at a steady incubation temperature of $T=60^{\circ}\text{C}$, as a function of the liquid base-to-curing agent ratio, r .

Figure R2.7. EDX analysis of samples resulting from the interaction of DI water and PDMS, after an exposition of $\sim 48h$ at a constant incubation temperature of $T_1 = 37^{\circ}\text{C}$, as a function of the liquid base:curing agent ratio r .

Figure R2.8. EDX analysis of samples resulting from the interaction of DI water and PDMS, after an exposition of $\sim 108h$ at a constant incubation temperature of $T_1 = 37^\circ\text{C}$, as a function of the liquid base:curing agent ratio r .

Figure R2.9. EDX analysis of samples resulting from the interaction of DI water and PDMS, after an exposition of $\sim 24h$ at a constant incubation temperature of $T_2 = 60^\circ\text{C}$, as a function of the liquid base:curing agent ratio r .

Figure R2.10. EDX analysis of samples resulting from the interaction of DI water and PDMS, after an exposition of $\sim 48h$ at a constant incubation temperature of $T_2 = 60^\circ\text{C}$, as a function of the liquid base:curing agent ratio r .

Figure R2.11. EDX analysis of samples resulting from the interaction of DI water and PDMS, after an exposition of $\sim 108h$ at a constant incubation temperature of $T_2 = 60^\circ\text{C}$, as a function of the liquid base:curing agent ratio r .

Results illustrate that the main elements found in the solution at contact with PDMS for prolonged amount of times are Carbon, Oxygen, and Silicon, that is consistent with preliminary analysis performed by Raman analysis and with independent reports (2, 3). In particular, Oxygen is, by far, the most abundant element in the PDMS residue, followed by Carbon and Silicon. At 37 °C (a temperature of interest since it is similar to that used in the cell-experiments reported in the work) none of the considered elements correlate with r . As an example, for a leakage time of 48 h, the relative abundance of Si varies from $\sim 9\%$ for $r = 7$, to $\sim 4\%$ for $r = 10$, to $\sim 16\%$ for $r = 13$. For a leaching time of 108 h, the content of silicon in the residual solute is of $\sim 5\%$ for $r = 7$, of $\sim 2.5\%$ for $r = 10$, of $\sim 2.7\%$ for $r = 13$.

Considering a higher value of temperature $T=60\text{ }^{\circ}\text{C}$, we observe that the abundance of Si decreases with r for 24 h and 48 h leaching times, while it oscillates with r between the 1.7 % and 2.2 % values for a 108 h leaching time.

The dependence of the relative abundance of Si on time and temperature is illustrated in **Figures R2.12** and **R2.13**.

Figure R2.12. Relative content of silicon in the solution resulting from the exposure of DI water with PDMS, as a function of *time*, for different values of the liquid base:curing agent ratio r .

Figure R2.13. Relative content of silicon in the solution resulting from the exposure of DI water with PDMS, as a function of *temperature*, for different values of the liquid base:curing agent ratio r .

Figure R2.12 shows that, for a fixed temperature $T=37\text{ }^{\circ}\text{C}$, the relative amount of Si in the final solution decreases linearly with time, with a slope that is relevant for $r=15$, less relevant and for $r=7$, and negligible

for $r=10$ (again: the rate of change of [Si] with T, does not correlate with r). Conversely, for a fixed temperature $T=60^{\circ}\text{C}$, the relative amount of Si in the final solution oscillates with time, and reaches a maximum at 48 h. Notably, the relative abundance of Si at 48 h is higher for $r=7$, followed by $r=10$, and by $r=15$. Thus, at 37°C and 60°C r seems to play different roles, enhancing (hampering) the release of Si at higher (lower) temperature.

Also the temperature (T) seems to have opposite effects on the content of Si, depending on the leaching time (**Figure R2.13**). At 48 h, [Si] increases with T. At 108 h, [Si] decreases with T. For both 48 and 108 h, the rate of change of [Si] with T is more relevant for $r = 7$, and less relevant for $r = 10$.

Thus, the results of Raman and EDX analyses, collectively, indicate that the leakage of PDMS may be less relevant than the mechanical properties of substrates in determining cell behavior, for reasons that can be summarized as follows:

1. The Raman signal of PDMS traces is, in any case, vanishingly small, and could be detected only through SERS effects. This indicates that leakage of PDMS is negligible.
2. The variation of Raman signals associated to Si-C stretching is small for varying values of r , which in turn indicates that the liquid-base:curing-agent ratio influences only moderately the leakage.
3. EDX analysis of samples illustrates that, in a given amount of PDMS excess, the relative abundance of Silicon correlates poorly with r – similarly to other elements found in solution. This indicates that, when present, the effects of leakage cannot explain the enhanced adhesion and enhanced clustering of neurons – that is instead related to the inverse of r .
4. Even assuming a significant release of Si into water or the culture medium used for neurons, silicon, in the form of silicon dioxide, or silicon-based nano- and micro-particles, and particulate, is generally considered to be biocompatible and not toxic to cells under many conditions (4, 5). The biocompatibility of Si-based nano- and micro-scale materials has been a focus of ongoing research efforts to understand the factors affecting their interactions with biological systems.

Considering all this, we confidently rule out that leakage is responsible for the peculiar cell behavior observed and reported in our work, either alone or combined with other mechanisms or PDMS characteristics, such as elasticity. These analyses and comments thereof have been conveniently inserted in the text and in a separate supporting information file.

References for section R2.1

(1) Fanse S, Bao Q, Zou Y, Wang Y, Burgess DJ. Effect of crosslinking on the physicochemical properties of polydimethylsiloxane-based levonorgestrel intrauterine systems. *Int J Pharm.* 2021 Nov 20;609:121192. doi: 10.1016/j.ijpharm.2021.121192. Epub 2021 Oct 16. PMID: 34666142; PMCID: PMC9236551..

(2) Regehr KJ, Domenech M, Koepsel JT, Carver KC, Ellison-Zelski SJ, Murphy WL, Schuler LA, Alarid ET, Beebe DJ. Biological implications of polydimethylsiloxane-based microfluidic cell culture. *Lab Chip.* 2009 Aug 7;9(15):2132-9. doi: 10.1039/b903043c. Epub 2009 Jun 4. PMID: 19606288; PMCID: PMC2792742.

(3) Sarah-Sophia D. Carter, Abdul-Raouf Atif, Sandeep Kadekar, Ingela Lanekoff, Håkan Engqvist, Oommen P. Varghese, Maria Tenje, Gemma Mestres, PDMS leaching and its implications for on-chip studies focusing on bone regeneration applications, *Organs-on-a-Chip*, Volume 2, 2020, 100004, ISSN 2666-1020, <https://doi.org/10.1016/j.ooc.2020.100004>.

(4) Sivakumar Murugadoss, Dominique Lison, Lode Godderis, Sybille Van Den Brule, Jan Mast, Frederic Brassinne, Noham Sebaihi, Peter H Hoet. Toxicology of silica nanoparticles: an update. *Arch Toxicol*. 91(9):2967-3010. 2017.

(5) Hamsa Jaganathana and Biana Godin. Biocompatibility Assessment of Si-based Nano- and Microparticles. *Adv Drug Deliv Rev*. 64(15): 1800–1819. 2012.

2. I am still concerned about authors claiming a linear relationship based on data that is inverse U-shaped (Figure 3, panels 'c' and 'f' in particular). What are the sample numbers and p-values for linear correlation coefficients authors are reporting? This is important because the inverse U-shape of the data suggests toxicity at extreme low and high base/curing agent ratios.

The p-values from the linear regression analyses examining the correlation between N and E (number of cells/elasticity) at various times post-seeding are 0.078 (24 hours), 0.123 (48 hours), 0.063 (72 hours), and 0.117 (96 hours). These values are low enough, even when taking into account the irregularities highlighted by the reviewer, to support our confidence in dismissing the possibility of an inverse U-shaped data pattern and toxicity effects due to PDMS leakage, as argued previously in response to point R2.1.

3. In the rebuttal, authors state that a decrease in cell numbers at 72 hrs and subsequent increase in cell numbers at 96 hrs is due unbalanced distribution of cells during cell seeding. This suggests a very high experimental or measurement variability – this experiment does not appear to have been conducted with scientific rigor. Authors did not address the concern about neuronal viability or glial cell proliferation: these processes can be concurrent (i.e. neurons could be dying, while glia could be dividing), resulting in unpredictable changes in overall cell number. In general, authors do not seem to be aware that primary hippocampal cultures contain proliferative (especially in 10% FBS) glial cells in addition to neurons. This is a major weakness of the manuscript.

The authors used a well-known standard method for collecting and culturing primary hippocampal neurons (6). Using such a protocol, as also underlined by the reviewer, it is possible to find both neurons and glial cells. Anyway, based on the experience and previous work of some of the authors (7-9), the proportion of glial cells compared to the overall number of cells present on the substrates' surfaces is relatively low.

As a proof of concept, several images (not shown in the manuscript) were taken and analyzed after 15 days. **Figure R2.14** shows the fluorescence acquired after the staining of cells with DAPI. Since this staining is not selective, the cells imaged in the figure can be either neurons or glial cells. However, **Figure R2.15** shows the same images but in another fluorescent channel, based on cells staining with anti-NeuN antibodies, specific for neurons' nuclei. By overlapping these two channels, it was possible to analyze the degree of colocalization between the nuclei of all the cells present in culture (**Figure R2.16a**) and hippocampal neurons (**b**). Before the colocalization process, cells were grayscale converted (**c**). Then, in

each image, the foreground and the background were separated by setting an automatic threshold (d). Following colocalization, the spatial overlap between the fluorescent labels associated with the cell and the neuron was determined (e). Colocalization results are described in terms of quantitative parameters, such as the Pearson correlation coefficient, the Manders M1 and M2 coefficients (f). The Manders colocalization coefficients are especially important. They describe the degree of overlap of the signal content of one channel (or image) to another (10-11). Thus, the values of M1 and M2 coefficients that we have found in this study, in fact, indicate that the great majority of neuronal cells overlap with the overall cell population (M1=0.965), and that of the cells imaged by nuclei-DAPI staining, about the 75% is represented by neurons (M2=0.74).

Therefore, these analyses, made on $n \sim 80$ images, prove that the great majority of cells (i.e., $\approx 75\%$) on the coverslips are represented by neurons. Notably, it is important to highlight that these images were acquired after 15 days in vitro, which is a much more extended period of time compared to the ones used for the network characterization (i.e., up to 4 days). This information is crucial as, unlike neurons, glial cells proliferate in time. Therefore, the presence of such a small component of glial cells after more than 2 weeks in culture clearly indicates that the data gathered and analyzed in previous experiments, which our conclusions are based on, can be related almost exclusively to neurons' contribution.

Further to this end, much of the results of this work are based on functional multi-calcium imaging (fMCI) experiments designed to examine cell activity as a function of PDMS preparation (presented, for example, in Figure 6 of the MS). Similar experiments are sensitive - by definition - to neurons solely, and are not affected under any circumstances by glia or other non-neuronal cells. Results of fMCI experiments, which show that neuronal activity correlates with material compliance, are consistent with the observed adhesion and clustering of neurons on PDMS – and support the research findings that in the low MPa range, substrate elasticity and neuronal performance are inversely correlated.

These comments have been conveniently inserted in the MS.

Figure R2.14. Fluorescence images of cells extracted from mouse hippocampus. Cell-nuclei were stained with DAPI.

Figure R2.15. Fluorescence images of neurons from the originating mouse hippocampal cell culture imaged in Figure R2.14.

Figure R2.16. Colocalization between the signal of fluorescence relative to mouse hippocampal cells (**a**) and hippocampal neurons (**b**). Before the colocalization process, cells were grayscale converted (**c**). Then, in each image the foreground and the background were separated by setting an automatic threshold (**d**). Following colocalization, the spatial overlap between the fluorescent labels associated to the cell and the neuron, was determined (**e**). Colocalization results are described in terms of quantitative parameters, such as the Pearson correlation coefficient, the Manders M1 and M2 coefficients (**f**).

References for section 2.3

- (6) Kaech, S. and G. Banker, Culturing hippocampal neurons. *Nat Protoc*, 2006. 1(5): p. 2406-15
- (7) Onesto V, Cancedda L, Coluccio ML, Nanni M, Pesce M, Malara N, Cesarelli M, Di Fabrizio E, Amato F, Gentile F. Nano-topography Enhances Communication in Neural Cells Networks. *Sci Rep*. 2017 Aug 29;7(1):9841. doi: 10.1038/s41598-017-09741-w.PMID: 28851984
- (8) Savardi A.; Borgogno M.; Narducci R.; La Sala G.; Ortega J. A.; Summa M.; Armirotti A.; Bertorelli R.; Contestabile A.; De Vivo M.; et al. Discovery of a Small Molecule Drug Candidate for Selective NKCC1 Inhibition in Brain Disorders. *Chem* 2020, 6 (8), 2073–2096. 10.1016/j.chempr.2020.06.017

- (9) Savardi A., Patricelli Malizia A., De Vivo M., Cancedda L., Borgogno M. Preclinical Development of the Na-K-2Cl Co-transporter-1 (NKCC1) Inhibitor ARN23746 for the Treatment of Neurodevelopmental Disorders *Pharmacol Transl Sci* 2023 Jan 4;6(1):1-11. doi: 10.1021/acsptsci.2c00197.
- (10) Jonathan W.D. Comeau, Santiago Costantino, Paul W. Wiseman. A Guide to Accurate Fluorescence Microscopy Colocalization Measurements. *Biophysical Journal*. Volume 91, Issue 12, Pages 4611-4622. 2006.
- (11) Kenneth W. Dunn, Malgorzata M. Kamocka, and John H. McDonald. A practical guide to evaluating colocalization in biological microscopy. *Biophysical Journal*. *Am J Physiol Cell Physiol*. 300(4): C723–C742. 2011.

4. Authors presented a new image-based method of assessing formation of networks in culture. Unfortunately, the method is not neuron specific. Phalloidin staining was used to label actin filaments – but this method can also label glial processes. There is no evidence that networks authors have derived from culture images are actually neuronal due to lack of neuron-specific staining markers.

We appreciate the observation of the Reviewer.

We apply the same logic from our previous discussion to tackle this issue. Neurons were not specifically stained in these experiments, and the networks determined after the physical connections between cells as imaged by fluorescence microscopy are not strictly representative of solely neuronal graphs.

Nonetheless, findings in response to point R2.3 reveal a minor presence of glial cells after more than two weeks in culture. This suggests that the topology of cell networks, as determined through both direct and indirect analysis, can be ascribed almost exclusively to neurons' contribution.

It's also important to mention that glial cells have various roles, including providing nutrients and oxygen to neurons—essentially, supporting signal transmission. Although they may not be directly involved in information processing, glial cells play a role in the overall signal exchange within the brain and nervous system. Therefore, a comprehensive evaluation of the architecture of hippocampal cells, such as that offered by calculating the small world coefficient, serves as an indicator of the brain's functional connectivity, its operational dynamics, and its overall performance.

5. Authors added description of the calcium imaging method.

We thank the Reviewer for the observation.

References.

Regehr KJ, Domenech M, Koepsel JT, Carver KC, Ellison-Zelski SJ, Murphy WL, Schuler LA, Alarid ET, Beebe DJ. Biological implications of polydimethylsiloxane-based microfluidic cell culture. *Lab Chip*. 2009 Aug 7;9(15):2132-9. doi: 10.1039/b903043c. Epub 2009 Jun 4. PMID: 19606288; PMCID: PMC2792742.

Sarah-Sophia D. Carter, Abdul-Raouf Atif, Sandeep Kadekar, Ingela Lanekoff, Håkan Engqvist, Oommen P. Varghese, Maria Tenje, Gemma Mestres, PDMS leaching and its implications for on-chip studies focusing on bone regeneration applications, *Organs-on-a-Chip*, Volume 2, 2020, 100004, ISSN 2666-1020, <https://doi.org/10.1016/j.ooc.2020.100004>.

Reviewer #3 (Remarks to the Author):

The authors have sufficiently resolved my concerns. I have no further comments regarding this manuscript. And I would like to thank the authors for their efforts in the revision.

We thank the Reviewer for the positive comments on the response and the MS.